# BET family members Bdf1/2 modulate global transcription initiation and elongation in *Saccharomyces cerevisiae*

**Rafal Donczew\*, Steven Hahn\***

Fred Hutchinson Cancer Research Center, Division of Basic Sciences, Seattle, United States

**Abstract** Human bromodomain and extra-terminal domain (BET) family members are promising targets for therapy of cancer and immunoinflammatory diseases, but their mechanisms of action and functional redundancies are poorly understood. Bdf1/2, yeast homologues of the human BET factors, were previously proposed to target transcription factor TFIID to acetylated histone H4, analogous to bromodomains that are present within the largest subunit of metazoan TFIID. We investigated the genome-wide roles of Bdf1/2 and found that their important contributions to transcription extend beyond TFIID function as transcription of many genes is more sensitive to Bdf1/2 than to TFIID depletion. Bdf1/2 co-occupy the majority of yeast promoters and affect preinitiation complex formation through recruitment of TFIID, Mediator, and basal transcription factors to chromatin. Surprisingly, we discovered that hypersensitivity of genes to Bdf1/2 depletion results from combined defects in transcription initiation and early elongation, a striking functional similarity to human BET proteins, most notably Brd4. Our results establish Bdf1/2 as critical for yeast transcription and provide important mechanistic insights into the function of BET proteins in all eukaryotes.

**\*For correspondence:**
rdonczew@fredhutch.org (RD);
shahn@fredhutch.org (SH)

**Competing interests:** The authors declare that no competing interests exist.

## Introduction

Bromodomains (BDs) are reader modules that allow protein targeting to chromatin via interactions with acetylated histone tails. BD-containing factors are usually involved in gene transcription, and their deregulation has been implicated in a spectrum of cancers and immunoinflammatory and neurological conditions (*Fujisawa and Filippakopoulos, 2017*; *Wang et al., 2021*). The bromodomain and extra-terminal domain (BET) family is characterized by the presence of a double BD, which has the highest affinity towards hyperacetylated histone H4, and an ET domain that interacts with non-histone proteins (*Rahman et al., 2011*; *Slaughter et al., 2021*). BET BD inhibitors have shown promising results in treating both blood cancers and solid tumors (*Stathis and Bertoni, 2018*). Early studies implicated mammalian BET proteins in regulation of selected lineage-specific genes (*Delmore et al., 2011*; *Lovén et al., 2013*); however, it has become clear that their regulatory roles extend to the majority of genes transcribed by RNA polymerase II (Pol II) (*Muhar et al., 2018*; *Winter et al., 2017*).

While most mammalian tissues express three BET factors (Brd2, Brd3, Brd4) (*Uhlén et al., 2015*), many studies have implicated Brd4 as most important for widespread changes in transcription after BET inactivation (*Muhar et al., 2018*; *Zheng et al., 2021*; *Zuber et al., 2011*). These broad genome-wide expression defects are due to the important roles of Brd4 in both transcription initiation and elongation. Brd4 contributes to recruitment of the transcription coactivator Mediator to enhancers and promoters and cooperates with Mediator in forming nuclear condensates at actively transcribed regions (*Bhagwat et al., 2016*; *Han et al., 2020*; *Sabari et al., 2018*). Brd4 is also important for productive transcription elongation, but it is unclear what mechanisms are involved.

**eLife digest** When a healthy cell creates new proteins, it activates a standard two-step biological manufacturing process. Firstly, DNA is transcribed from a specific gene to generate a strand of messenger RNA, or mRNA. Next, this mRNA molecule is translated to create the final protein product. This process of converting DNA into mRNA is supported by a series of helper proteins, including proteins from the bromodomain and extra-terminal domain (BET) family.

Cancer cells can become 'addicted' to the process of converting DNA into RNA, leading to the overproduction of mRNA molecules, uncontrolled cell growth and tumor formation. Knocking out BET helper proteins could potentially bring cancer cells under control by halting transcription and preventing tumor growth. However, the precise ways in which BET helper proteins regulate transcription are currently poorly understood, and therefore developing rational ways to target them is a challenge.

Building on their previous work, Donczew and Hahn have investigated how two BET helper proteins, Bdf1 and Bdf2, help to regulate transcription in budding yeast. Using a range of genomic techniques, Donczew and Hahn found that Bdf1 and Bdf2 had important roles for initiating transcription and elongating mRNA molecules. Both BET proteins were also involved in recruiting other protein factors to help with the transcription process, including TFIID and Mediator. Based on these findings, it is likely that cooperation between BET proteins, TFIID and Mediator represents a common pathway through which gene expression is regulated across all eukaryotic organisms.

Both Bdf1 and Bdf2 were also found to provide the same functions in yeast as similar BET proteins in humans. Using this robust yeast model system to perform further detailed studies of BET factors could therefore provide highly relevant information to expand our understanding of human biology and disease.

Ultimately, this research provides important insights into how two members of the BET family of helper proteins contribute to the control of transcription in yeast. This information could be used to guide the design of new drugs for cancer therapy that target not only BET proteins themselves but also other proteins they recruit, including TFIID and Mediator. Such targeted drugs would be expected to be more harmful for cancer cells than for healthy cells, which could reduce unwanted side effects.

Brd4 associates with the positive transcription elongation factor b (P-TEFb), but no global defects in P-TEFb recruitment to chromatin were observed following inactivation of Brd4 or all BET factors (*Muhar et al., 2018*; *Winter et al., 2017*). Proposed alternative mechanisms of elongation control include release of P-TEFb inhibition, histone chaperone activity, or direct kinase activity (*Devaiah et al., 2012*; *Itzen et al., 2014*; *Kanno et al., 2014*). Finally, Brd4 was shown to affect preinitiation complex (PIC) formation at promoters, although its roles in later stages of transcription are believed to be dominant (*Kanno et al., 2014*; *Winter et al., 2017*).

TFIID is a conserved Pol II factor that, in yeast, regulates transcription of almost all Pol II-transcribed genes (*Donczew and Hahn, 2018*; *Huisinga and Pugh, 2004*; *Warfield et al., 2017*). TFIID comprises TBP (TATA binding protein) and 13–14 Tafs (TBP-associated factors). TFIID acts as a TBP-DNA loading factor, a promoter recognition factor and a molecular scaffold guiding PIC assembly (*Patel et al., 2020*). Recruitment of metazoan TFIID to promoters is thought to be aided by interactions between two Taf subunits and chromatin marks that are enriched at the +1 nucleosome. The human Taf1 double BD and Taf3 PHD domain recognize acetylated histone H4 tails and trimethylated histone H3 lysine 4, respectively (*Jacobson et al., 2000*; *van Ingen et al., 2008*). In contrast, Taf1 and Taf3 in budding yeast are missing both of these chromatin readers as well as recognizable promoter DNA sequence motifs such as the INR and DPE that are known to interact with metazoan Tafs (*Patel et al., 2020*).

Yeast BET family member Bdf1 and its paralogue Bdf2 are genetically redundant, with at least one required for viability. Deletion of *bdf1* but not *bdf2* results in growth retardation, suggesting a dominant role for Bdf1 (*Matangkasombut et al., 2000*). Bdf1 binds Taf7 and recognizes acetylated histone H4. Both of these functions were found to support cell growth and transcription at several loci. Based on these findings, it was proposed that Bdf1 double BD substitutes for the missing BDs

of yeast Taf1, linking yeast TFIID to promoter-enriched chromatin marks (*Matangkasombut et al., 2000*; *Matangkasombut and Buratowski, 2003*). Gene deletion or targeted depletion of Bdf1 revealed a small contribution of Bdf1 to Taf1 recruitment genome-wide with a bias towards a group of genes classified as 'TFIID-dominated' and led to a modest defect in transcription at a limited gene set (*Durant and Pugh, 2007*; *Joo et al., 2017*; *Ladurner et al., 2003*).

By applying degron-mediated protein depletion and monitoring nascent mRNA levels, we recently showed that yeast protein-coding genes can be classified into two broad categories based on transcription changes after rapid depletion of the coactivators TFIID and SAGA (*Donczew et al., 2020*). We found that the majority of yeast genes are strictly TFIID-dependent while a subset, which we termed coactivator-redundant (CR) genes, are co-regulated by TFIID and SAGA. This latter category overlaps ~50% with the gene set earlier termed 'SAGA-dominated' (*Huisinga and Pugh, 2004*). In contrast, long-term ablation of SAGA via gene deletions showed that nearly all yeast genes are dependent on chromatin modifications directed by SAGA. However, the molecular basis for the TFIID and CR gene classes remains unknown. For example, it is not known if there are other factors that preferentially participate in the TFIID or SAGA-directed pathways.

Here, we have examined the functions of Bdf1 and Bdf2 in yeast transcription and found broad genome-wide roles for the Bdfs in both transcription initiation and elongation. Rapid Bdf depletion strongly decreases transcription from the TFIID-dependent genes, and, at many genes, the Bdfs are more important than Tafs for normal levels of transcription. The Bdfs contribute to PIC formation and TFIID and Mediator recruitment to gene regulatory regions. In addition, at many genes, the Bdfs also regulate transcription elongation and this role contributes to the strong defects in mRNA synthesis upon Bdf depletion. These striking functional similarities with mammalian BET factors suggest broad conservation of BET function in eukaryotes.

## Results

### Transcription of most TFIID-dependent genes is more sensitive to Bdf1/2 than to TFIID depletion

A limitation of previous studies on Bdf1/2 function was that both proteins could not be simultaneously eliminated since at least one Bdf is required for viability. In this study, we used the auxin-degron system (*Nishimura et al., 2009*) to achieve rapid depletion of Bdf1, Bdf2, or simultaneous depletion of both proteins, and used 4-thioU RNA-seq to monitor changes in newly synthesized mRNA (*Donczew et al., 2020*; *Rabani et al., 2011*). We first assessed protein degradation after a 30 min treatment with the auxin indole-3-acetic acid (IAA) (*Figure 1—figure supplement 1A*). Bdf1 was efficiently degraded with <10% of protein remaining in IAA-treated samples. Bdf2 degradation was less complete, with ~15% of protein remaining, so we prepared a *bdf2* deletion strain combined with a Bdf1 degron (Bdf1, Δ*bdf2*). Growth of all strains with Bdf derivatives was similar to wild type (*Figure 1—figure supplement 1B*), and analysis of 4-thioU mRNA levels in control dimethyl sulfoxide (DMSO)-treated cultures showed that changes in transcription due to these genetic alterations were minimal (*Figure 1—figure supplement 1C*).

Experiments were done in two or three biological replicates, and the variation in nascent mRNA levels between replicate samples was <30% for nearly all experiments (*Figure 1—figure supplement 1D* and *Supplementary file 1*). Bdf2 degradation or *bdf2* deletion showed minimal changes in transcription while Bdf1 degradation resulted in modest defects, in agreement with the genetic phenotypes of *BDF1/2* mutants (*Figure 1A*, *Figure 1—figure supplement 1E*, and *Supplementary file 1*). Strikingly, depletion of both Bdf1 and Bdf2, either through a double degron or by combining a *bdf2* deletion with a Bdf1 degron, resulted in a global collapse of transcription with median decreases of ~4-fold and ~5.4-fold, respectively (*Figure 1A*, *Figure 1—figure supplement 1F*, and *Supplementary file 1*). Transcriptional changes in both Bdf1/2 depletion experiments are highly correlated ($r = 0.95$) (*Figure 1—figure supplement 1G*). The results of the Bdf1 and Bdf1/2 depletion experiments also correlate ($r = 0.82$), showing redundant roles of Bdf1 and Bdf2 in transcription (*Figure 1—figure supplement 1H*).

We compared results of the Bdf depletion experiment with previously published results of rapid Taf1 depletion (*Donczew et al., 2020*). Surprisingly, we observed that transcription changes upon Bdf or Taf1 depletion correlate but the relationship between them is not linear (*Figure 1B*). Labeling

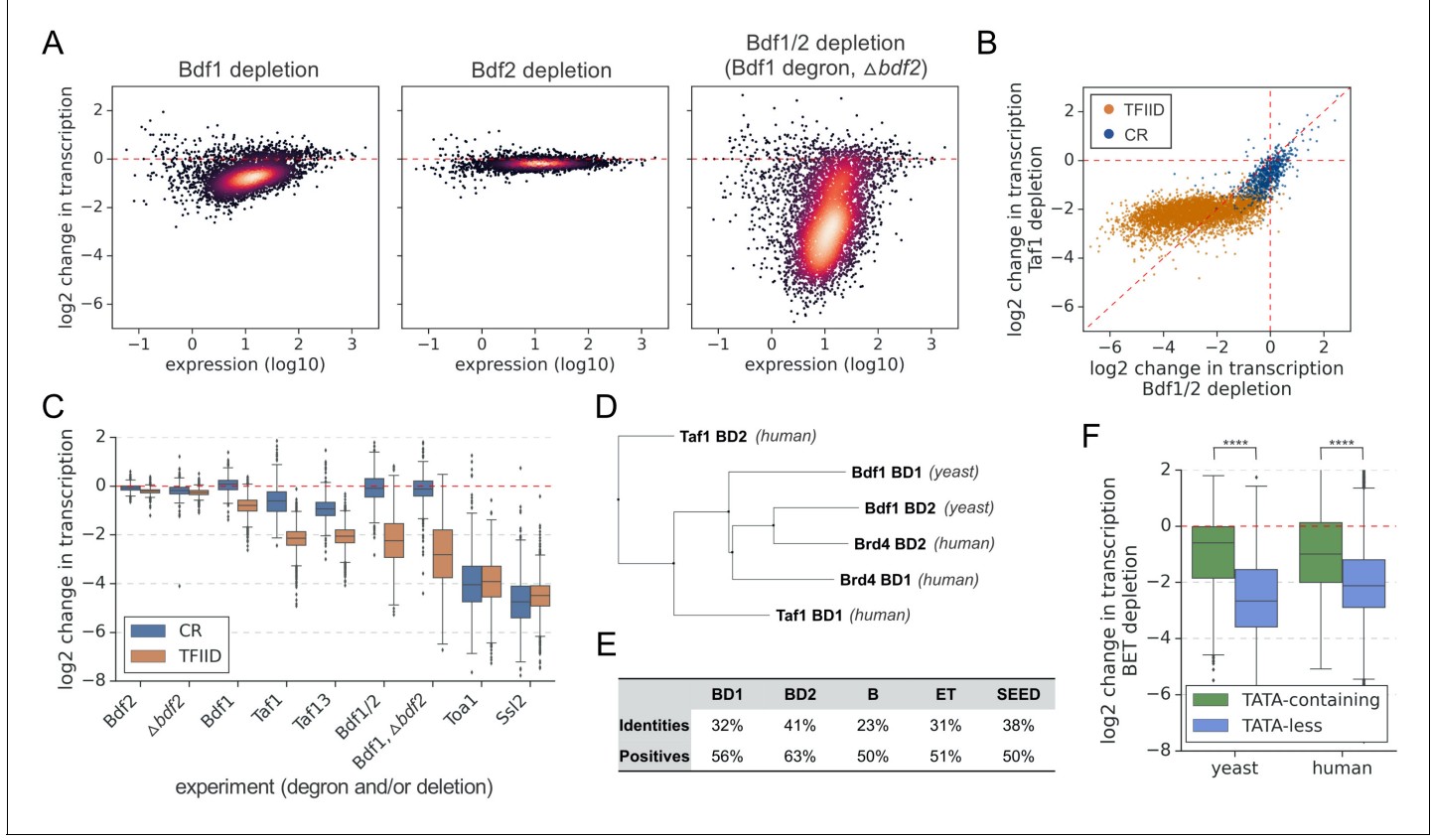

**Figure 1.** Bdf1/2 are critical for transcription of TFIID-dependent genes and share significant similarities with human bromodomain and extra-terminal domain (BET) factors. (A) Transcriptional changes caused by degron depletion of Bdf1/2 for 30 min. Scatter plots comparing gene expression level (in log scale) with $\log_2$ change in transcription per gene measured by 4-thioU RNA-seq. Mean values from replicate experiments are plotted for 5313 genes with detectable signals in all RNA-seq samples collected in this work. (B) Scatter plot comparing $\log_2$ change in transcription after depleting Bdf1/2 (this work) or Taf1 (*Donczew et al., 2020*). Data for 4883 genes previously classified into TFIID-dependent and coactivator-redundant (CR) categories are shown (*Donczew et al., 2020*). (C) Boxplot showing $\log_2$ change in transcription for 4883 genes measured by 4-thioU RNA-seq after depleting indicated factors. Genes are grouped into TFIID-dependent and CR categories. (D) Phylogenetic analysis of amino acid sequences of individual bromodomains from human Taf1, human Brd4, and yeast Bdf1. (E) BLAST global alignment of amino acid sequences within indicated domains of Bdf1 and Brd4. (F) Boxplot showing $\log_2$ change in transcription after depleting BET factors in yeast (this work) or human cells (*Winter et al., 2017*). Genes are classified depending on the presence of a consensus TATA box (TATAWAW) in their promoter. A list of TATA-containing promoters in human cells was obtained from Eukaryotic Promoter Database (EPD). A single, most representative promoter per gene was used in this analysis. Results of the Welch's t-test are shown. The asterisks represent p-value with the following cutoff levels: ****0.0001, *** 0.001, ** 0.01, * 0.05, ns = > 0.05. See also *Supplementary files 1* and *2*.

The online version of this article includes the following source data and figure supplement(s) for figure 1:

**Figure supplement 1.** Bdf1/2 are critical for transcription of TFIID-dependent genes.

**Figure supplement 1—source data 1.** Original, unedited image of a western blot for the Bdf1 depletion experiment (A).

**Figure supplement 1—source data 2.** Original, unedited image of a western blot for the Bdf1 depletion experiment (A).

**Figure supplement 1—source data 3.** Original, unedited images of a western blot for the Bdf2 depletion experiment (A).

**Figure supplement 1—source data 4.** Original, unedited images of a western blot for the Bdf1/2 depletion experiment (A).

**Figure supplement 1—source data 5.** Uncropped images of western blots shown in (A) with the relevant cropped bands marked with red rectangles.

**Figure supplement 1—source data 6.** Original image of a YPD plate for spot assay analysis (B).

**Figure supplement 1—source data 7.** Original image of a YPD plate for spot assay analysis (B).

**Figure supplement 1—source data 8.** Uncropped images of plates shown in (B) with the relevant cropped area marked with a red rectangle.

**Figure supplement 2.** Bdf1 and Tafs are degraded with similar efficiency.

**Figure supplement 2—source data 1.** Original, unedited image of a western blot for the experiment comparing the degradation of Bdf1, Taf1, and Taf13 (A).

**Figure supplement 2—source data 2.** Original, unedited image of a western blot for the experiment comparing the degradation of Bdf1, Taf1, and Taf13 (A).

**Figure supplement 2—source data 3.** Uncropped images of western blots shown in (A) with the relevant cropped bands marked with red rectangles.

*Figure 1 continued on next page*

*Figure 1 continued*

**Figure supplement 2—source data 4.** Original, unedited image of a western blot for the H2A.Z depletion experiment (D).

**Figure supplement 2—source data 5.** Uncropped image of a western blot shown in (D) with the relevant cropped bands marked with a red rectangle.

**Figure supplement 3.** Bdf1/2 share significant similarities with human bromodomain and extra-terminal domain (BET) factors.

the points based on the TFIID and CR gene categories revealed that genes less sensitive to Bdf depletion (on the right of the diagonal line) include almost all CR genes and one-third of TFIID-dependent genes. Interestingly, most of the TFIID-dependent genes (68%) are more sensitive to Bdf than Taf1 depletion and, for 12% of them, the difference is fourfold or higher (*Supplementary file 1*). Importantly, Bdf1 and Taf1 are degraded with similar efficiency (*Figure 1—figure supplement 2A*). Our results show that, while Bdfs likely act in conjunction with TFIID at many promoters, Bdfs also have important TFIID-independent contributions to transcription of many genes. Comparison of the Bdf results with depletion of Taf13, another TFIID subunit, leads to similar conclusions (*Figure 1—figure supplement 2A, B*).

Ribosomal protein (RP) genes are frequently analyzed as a separate gene category due to high expression levels, common regulatory mechanisms, and important roles in cell growth and stress response (*Zencir et al., 2020*). About 95% of RP genes belong to the TFIID-dependent class (*Donczew et al., 2020*; *Huisinga and Pugh, 2004*). Surprisingly, we found that nearly all RP genes are significantly less dependent on Bdf1/2 compared with TFIID, an uncommon feature for TFIID-dependent genes (*Figure 1—figure supplement 2C*).

With many TFIID-dependent genes losing >85% of detectable transcription after Bdf1/2 depletion, we investigated how these changes compare to defects caused by depletion of basal components of the PIC. We depleted basal factors with degron tags on either TFIIA subunit Toa1 or the TFIIH translocase subunit Ssl2 followed by 4-thioU RNA-seq. The results are summarized in a boxplot divided into CR and TFIID-dependent genes (*Figure 1C* and *Supplementary file 1*). Depletion of Toa1 or Ssl2 decreases all mRNA transcription independent of gene class with median changes of ~15-fold and ~23-fold, respectively. Interestingly, 38 and 19% of TFIID-dependent genes are similarly sensitive to Toa1 or Ssl2 depletion as they are to Bdf depletion. Combined, our results show that Bdf dependence is as good or even a better classifier for the TFIID and CR gene classes than Taf-dependence. Transcription at many TFIID-dependent genes is severely compromised in the absence of Bdfs while transcription of CR genes is only weakly affected (*Figure 1C*).

Bdf1 is also a subunit of the SWR1 complex that incorporates the histone variant H2A.Z into promoter-proximal nucleosomes (*Krogan et al., 2003*; *Zhang et al., 2005*). We used the degron system to rapidly deplete H2A.Z and check if the loss of transcription after Bdf depletion is mediated by defects in H2A.Z deposition (*Figure 1—figure supplement 2D*). RNA-seq analysis did not reveal significant defects in gene transcription following the rapid loss of H2A.Z (*Figure 1—figure supplement 2E* and *Supplementary file 1*), suggesting that the global transcriptional defects upon Bdf depletion are not mediated via SWR1 and H2A.Z.

## Bdf1/2 share significant similarities with human BET factors

Bdf1 was initially proposed to constitute a missing part of yeast Taf1 (*Matangkasombut et al., 2000*), but later work classified Bdf1/2 as members of the BET family based on a conserved domain organization (*Wu and Chiang, 2007*; *Figure 1—figure supplement 3A*). Since Bdfs have important TFIID-independent functions at many genes, we explored similarities between yeast and human BET proteins. Multiple sequence alignment and phylogenetic analysis of individual BDs of Bdf1, Brd4, and human Taf1 (hTaf1) revealed that Bdf1 BDs, especially BD2, are more closely related to Brd4 than hTaf1 BDs (*Figure 1D* and *Figure 1—figure supplement 3B*). Other conserved domains of Bdf1 and Brd4 show ≥50% similarity (*Figure 1E*). We next compared gene-specific responses to Bdf/BET depletion in yeast and human cells. We used a published RNA-seq dataset from an experiment where BET factors were targeted with the chemical degrader dBET6 in the MOLT4 leukemia cell line (*Winter et al., 2017*; *Supplementary file 1*). Since the CR/TFIID-dependent classes are biased for TATA-containing/TATA-less categories (*Donczew et al., 2020*; *Huisinga and Pugh, 2004*), we divided yeast and human genes into two categories based on the presence or absence of a consensus TATA box. We found that TATA-less genes in both systems were significantly more

affected by Bdf/BET depletion than TATA-containing genes (*Figure 1F*). Finally, in the 25% most affected genes in each dataset, we observed an extensive overlap in major gene ontology terms (*Figure 1—figure supplement 3C* and *Supplementary file 2*). Altogether, these results suggest a common biological role of BET factors in yeast and human cells, a conclusion supported by other results presented below.

## Bdf1/2 and Taf1 have similar genome-wide binding patterns

We recently mapped Taf1 and other TFIID subunits using an improved ChEC-seq method, where DNA cleavage by protein-MNase fusions is applied to map genome-wide binding (*Donczew et al., 2021*; *Donczew et al., 2020*). TFIID is detectable at over 3000 yeast promoters and is found at both TFIID-dependent and CR genes, which agrees with its general role in transcriptional regulation. We used ChEC-seq to map Bdf1 and Bdf2 binding genome-wide. Comparison of Bdf1/2 and Taf1 ChEC-seq showed similar binding patterns with broad peak spanning between −1 and +1

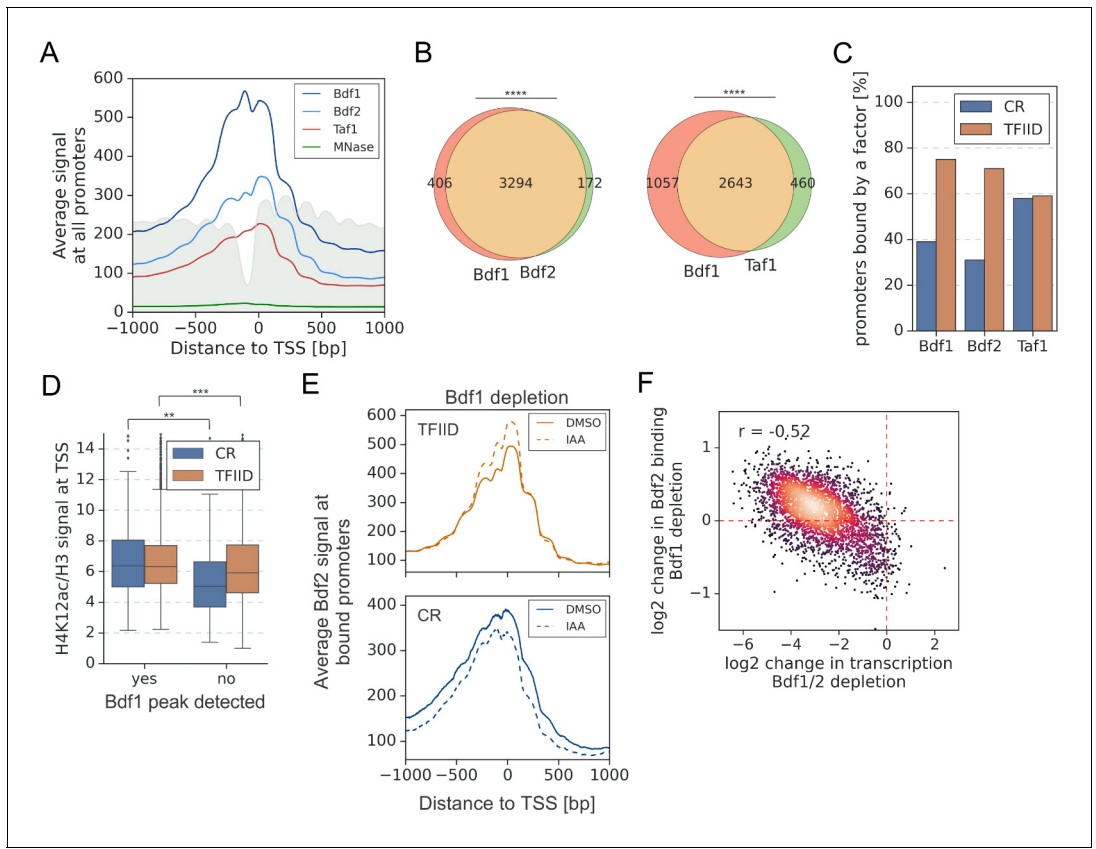

**Figure 2.** A significant overlap between Bdf1/2 and Taf1 bound promoters. (**A**) Average plot of Bdf1, Bdf2, and Taf1 ChEC-seq signals versus free MNase signal at 5888 yeast promoters. Published MNase-seq data are shown as a gray area plot (*Oberbeckmann et al., 2019*). Mean values from replicate experiments are plotted. (**B**) Venn diagrams showing the overlap of promoters bound by indicated factors. Results of the hypergeometric test are shown. The asterisks represent p-value with the following cutoff levels: **** 0.0001, ***0.001, ** 0.01, * 0.05, ns = >0.05. (**C**) Bar plot showing percentage of promoters in each class (TFIID-dependent or coactivator-redundant [CR]) bound by a given factor. (**D**) Boxplot showing the H4K12ac signal at transcription start site (TSS) normalized to H3 signal. Genes are grouped into TFIID-dependent and CR categories, and data are plotted separately depending on the presence of a significant Bdf1 peak. Signals were calculated in a −100 to +200 bp window relative to TSS. Published H3 ChIP-seq dataset was used for this analysis (*Bruzzone et al., 2018*). Results of the Welch's t-test are shown. (**E**) Average plot comparing Bdf2 ChEC-seq signal before (dimethyl sulfoxide [DMSO], solid line) and after (indole-3-acetic acid [IAA], dashed line) Bdf1 depletion at 3223 promoters bound by Bdf2 and classified into TFIID-dependent and CR categories. (**F**) Scatter plot comparing log₂ change in transcription and log₂ change in Bdf2 ChEC-seq occupancy after depleting Bdf1/2 or Bdf1, respectively. Spearman correlation coefficient (*r*) is shown. Bdf2 signal was calculated in a 200 bp window centered on a dominant peak assigned to 3223 promoters. See also *Supplementary file 3*.

The online version of this article includes the following figure supplement(s) for figure 2:

**Figure supplement 1.** A significant overlap between Bdf1/2 and Taf1 bound promoters.

nucleosomes and signals extending into the gene body, especially for Bdf (*Figure 2A* and *Figure 2—figure supplement 1A*). We identified 3700, 3466, and 3103 promoters bound by Bdf1, Bdf2, and Taf1, respectively. Data for Bdf1 and Bdf2 confirm redundancy between both factors. There is an extensive overlap of promoters bound by both Bdfs and the signals at common bound promoters are highly correlated, although Bdf1 signals are consistently stronger, in agreement with its dominant role (*Figure 2B*, *Figure 2—figure supplement 1B*, and *Supplementary file 3*). There is also extensive overlap between Bdf1 and Taf1 bound promoters (*Figure 2B*), although Bdf1 is found less frequently at CR genes (*Figure 2C*), consistent with the lesser role of Bdfs at this gene class. Interestingly, the Bdf1 chromatin occupancy is only weakly predictive of the gene dependence on Bdf1/2, as was similarly reported for human BET factors (*Figure 2—figure supplement 1C*; *Muhar et al., 2018*; *Winter et al., 2017*). We validated Bdf1 binding using ChIP-seq as an orthogonal approach. The results are similar to ChEC-seq as well as to previously published ChIP-exo data (*Figure 2—figure supplement 1D* and *Supplementary file 3*; *Rhee and Pugh, 2012*). The number of bound promoters detected by ChIP-seq is lower due to worse signal to background ratio but shows a good overlap with ChEC-seq results (*Figure 2—figure supplement 1E*). We also performed a Bdf2 ChIP-seq experiment, but the quality of the data was too low to call peaks. Instead, we calculated Bdf2 signal at promoters bound by Bdf1 and found that the signals correlate well, supporting conclusions obtained by ChEC-seq (*Figure 2—figure supplement 1B, F* and *Supplementary file 3*). However, a major difference between ChEC-seq and ChIP-seq was observed at the RP genes. We detected Bdf1 binding at 15% of RP promoters using ChEC-seq, which is consistent with a limited role of Bdfs at RP genes. Conversely, almost all (94%) RP promoters have a significant Bdf1 ChIP signal, but this result is not predictive of Bdf1 function (*Figure 1—figure supplement 2C* and *Figure 2—figure supplement 1G*).

Next, we investigated whether Bdf1 binding correlates to histone H4 acetylation levels. We measured the genome-wide acetylation of H4 lysine 12 (H4K12ac), which was shown to be a preferred target for Bdf1 (*Slaughter et al., 2021*; *Figure 2—figure supplement 1H*). Importantly, for both classes of genes, the normalized H4K12ac level at promoters significantly differs depending on the presence of a Bdf1 peak (*Figure 2D* and *Supplementary file 3*). To conclude, at many promoters Bdf1 occupancy predicts higher H4K12ac levels and transcription dependence on Bdfs.

It was earlier proposed that Bdf2 redistributes to preferred Bdf1 sites in a *bdf1* deletion strain (*Durant and Pugh, 2007*). We tested this hypothesis by measuring changes in Bdf1 or Bdf2 occupancy after depleting Bdf2 or Bdf1, respectively. Following Bdf1 depletion, the Bdf2 signal increases at TFIID promoters at the expense of CR promoters (*Figure 2E*). The changes at individual promoters are relatively weak but they correlate well with gene dependence on Bdfs ($r = -0.52$) (*Figure 2F* and *Supplementary file 3*). Conversely, Bdf1 does not redistribute in response to Bdf2 depletion (*Figure 2—figure supplement 1I* and *Supplementary file 3*). This result provides more evidence of redundancy between Bdf1 and Bdf2, with Bdf1 being the dominant factor and Bdf2 serving an auxiliary role.

## Bdf1 can be targeted to promoters independently of histone H4 acetylation

Esa1, a catalytic subunit of the NuA4 complex, is almost exclusively responsible for acetylation of lysines 5, 8, and 12 on histone H4 in yeast (*Chang and Pillus, 2009*; *Suka et al., 2001*). We compared the roles of Esa1 and Bdfs in transcriptional regulation using an Esa1 degron strain. We used a 1 hr treatment with IAA, which allowed for a substantial loss of both H4K12ac and Esa1 (*Figure 3—figure supplement 1A*). 4-thioU RNA-seq analysis revealed a global loss of transcription after Esa1 depletion, which has a good but nonlinear correlation with the Bdf experiment ($r = 0.73$) (*Figure 3A, B*, *Figure 3—figure supplement 1B*, and *Supplementary file 1*). Importantly, Bdf depletion results in stronger transcriptional defects, suggesting that a portion of Bdf function is H4Ac independent. Our findings show that Bdf1/2, Esa1, and TFIID cooperate in regulation of many genes, but their contributions to transcription can be significantly different at individual genes (*Figure 3C*, *Figure 3—figure supplement 1C*, and *Supplementary file 1*).

Bdf1 interactions with chromatin were previously shown to be modulated by Esa1 (*Durant and Pugh, 2007*; *Koerber et al., 2009*). We used ChEC-seq to measure changes in Bdf1 occupancy following Esa1 depletion. In parallel, we measured changes in H4K12ac level using ChIP-seq. After 1 hr of IAA treatment, the H4K12ac chromatin signal decreased dramatically, but Bdf1 was still bound at

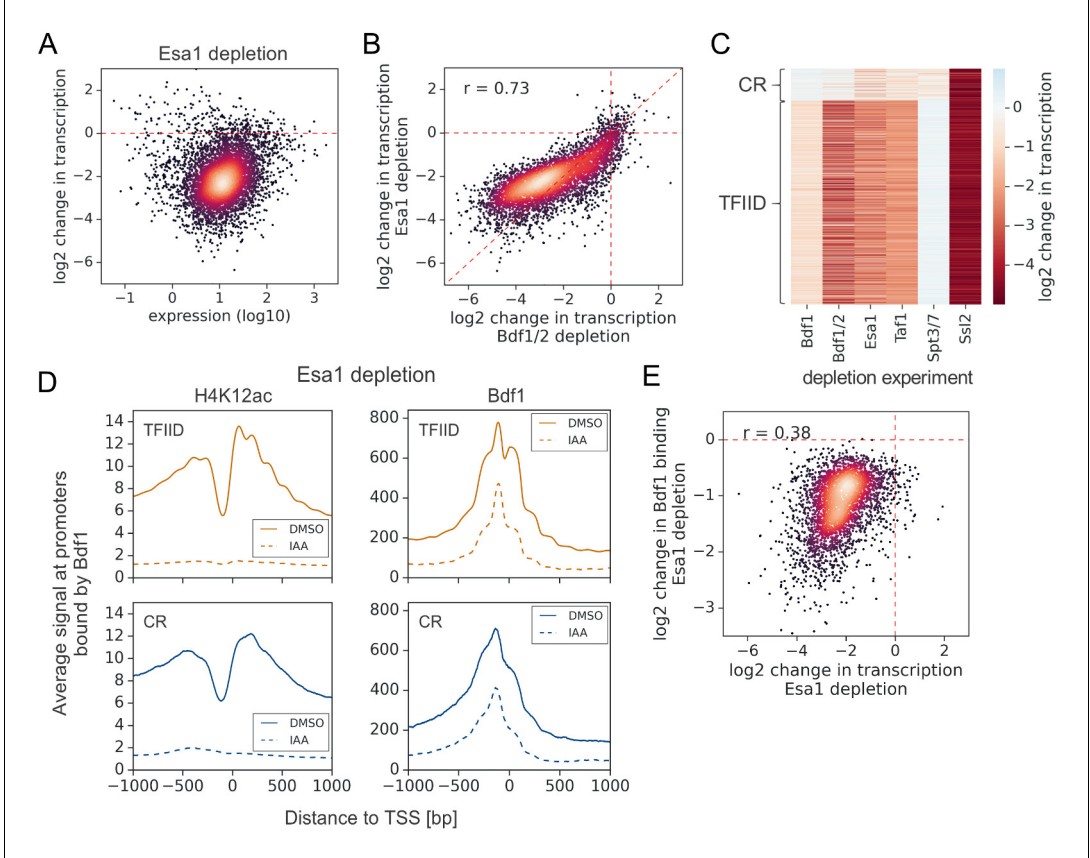

**Figure 3.** Esa1 regulates similar set of genes as Bdf1/2 but is not essential for Bdf1 targeting to chromatin. (**A**) Scatter plot comparing gene expression level (in log scale) with $log_2$ change in transcription per gene after depleting Esa1 for 60 min. Mean values from replicate experiments are plotted for 5313 genes with detectable signals in all RNA-seq samples collected in this work. (**B**) Scatter plot comparing $log_2$ change in transcription after depleting Bdf1/2 or Esa1. Spearman correlation coefficient (*r*) is shown. (**C**) Heatmap comparing results of indicated depletion experiments on transcription of 4883 genes. (**D**) Average plots comparing H4K12ac ChIP-seq and Bdf1 ChEC-seq signals before (dimethyl sulfoxide [DMSO], solid line) and after (indole-3-acetic acid [IAA], dashed line) Esa1 depletion at 3426 promoters bound by Bdf1 and classified into TFIID-dependent and coactivator-redundant (CR) categories. Mean values from replicate experiments are plotted. (**E**) Scatter plot comparing $log_2$ change in transcription and $log_2$ change in Bdf1 occupancy after depleting Esa1 at 3426 promoters. Spearman correlation coefficient (*r*) is shown. Bdf1 signal was calculated in a 200 bp window centered on a dominant peak. See also *Supplementary file 3*.

The online version of this article includes the following source data and figure supplement(s) for figure 3:

**Figure supplement 1.** Esa1 regulates similar set of genes as Bdf1/2 but is not essential for Bdf1 targeting to chromatin.

**Figure supplement 1—source data 1.** Original, unedited image of a western blot for the Esa1 depletion experiment (**A**).

**Figure supplement 1—source data 2.** Original, unedited image of a western blot for the Esa1 depletion experiment (**A**).

**Figure supplement 1—source data 3.** Uncropped images of western blots shown in (**A**) with the relevant cropped bands marked with red rectangles.

substantial levels, especially at promoters (*Figure 3D*, *Figure 3—figure supplement 1D*, and *Supplementary file 3*). Nevertheless, the quantified changes in Bdf1 promoter signal and transcription following Esa1 depletion are correlated, suggesting that the effect of Esa1 on transcription is largely mediated by Bdf1 (*Figure 3E*). Consistent with our results, Brd4 was also detected at human promoters following BD inhibition or mutation, which explains why BET degradation outperforms BET inhibition in preclinical cancer models (*Bauer et al., 2021*; *Kanno et al., 2014*; *Winter et al., 2017*). It is possible that the residual binding of Bdfs in the absence of H4 acetylation is due to interactions with other histone acetylation marks. For example, Bdf1 was shown to have weak affinity for acetylated histone H3 (*Matangkasombut and Buratowski, 2003*). It has also been proposed that BET recruitment can be mediated by interactions with transcription factors (TFs), coactivators, or chromatin remodelers, and it will be interesting to investigate if such mechanisms play a role in Bdf targeting (*Lambert et al., 2019*; *Rahman et al., 2011*; *Shen et al., 2015*).

## Bdf1/2 play a role in both TFIID and Mediator recruitment

Human BET factors are involved in recruitment of Mediator (*Bhagwat et al., 2016*), and Bdf1/2 were proposed to assist in recruitment of TFIID to yeast promoters (*Matangkasombut et al., 2000*). We used ChEC-seq to measure changes in chromatin occupancy of selected subunits of TFIID (Taf1, Taf11), Mediator (Med8, Med17), and SAGA (Spt3) after Bdf depletion. We first defined promoters bound by each factor and restricted the downstream analysis only to these locations. Different factors mapped in this study had characteristic distributions between the two classes of yeast promoters. Bdf1/2 are enriched at TFIID-dependent promoters, TFIID does not exhibit preference against either class, and SAGA and Mediator are enriched at CR promoters (*Figure 4—figure supplement 1A* and *Supplementary file 3*). We observed modest (~1.7-fold) decreases in TFIID and Mediator occupancy at all bound promoters following Bdf depletion (*Figure 4A, B*, *Figure 4—figure supplement 1B, C*, and *Supplementary file 3*). In contrast, SAGA binding was relatively insensitive to Bdf depletion with only small decreases observed, mostly at CR promoters. Importantly, changes measured for different subunits of TFIID or Mediator correlate (*Figure 4—figure supplement 1D, E* and *Supplementary file 3*). Using the ChIP-seq assay, similar results were observed for Bdf-dependent Taf1 binding (*Figure 4—figure supplement 1C, F* and *Supplementary file 3*).

We next tested whether the decreases in TFIID and Mediator recruitment were due to cooperative interactions between these two factors (*Grünberg et al., 2016*; *Knoll et al., 2018*) or via a more direct role of Bdfs in modulating TFIID and Mediator binding. We used ChEC-seq to measure changes in Taf1 and Med8 promoter signals after depleting Med14 and Taf13, respectively. We confirmed modest binding cooperativity between both factors, but the decreases in TFIID and Mediator binding were significantly stronger after depleting Bdfs (compare *Figure 4A, B* with C, D; *Supplementary file 3*). This indicates a role of Bdfs in both TFIID and Mediator recruitment beyond the cooperative interactions of these two factors. Altogether, our results illustrate that Bdfs have independent contributions to the recruitment of TFIID and Mediator, although, in the absence of Bdfs, both factors are still recruited to promoters at substantial levels. Our results suggest that the strong transcription defects upon Bdf depletion are not solely mediated by defects in TFIID and Mediator binding.

## PIC assembly, especially at TFIID-dependent genes, is strongly dependent on Bdf1/2

Since the strong decrease in transcription of TFIID-dependent genes could not be explained solely by decreases in TFIID or Mediator binding, we probed for defects in PIC assembly using ChIP-seq to quantitate promoter binding of the basal factor TFIIB. Depletion of either Bdfs or Taf13 caused a global loss of TFIIB, most pronounced at TFIID-dependent promoters (*Figure 5A*, *Figure 5—figure supplement 1*, and *Supplementary file 3*). Comparing the defects in TFIIB binding due to Bdf depletion versus TFIID depletion, we found that Bdf depletion caused a significantly larger loss of TFIIB at TFIID-dependent promoters than does TFIID depletion (*Figure 5B* and *Supplementary file 3*). Conversely, at CR genes, we found that TFIID depletion caused a greater loss in TFIIB signal compared with Bdf depletion. These results are consistent with the above RNA-seq experiments. However, comparison of transcriptional and TFIIB binding defects caused by Bdf depletion shows that Bdfs have a broader role in transcription than solely regulating PIC assembly. This larger role of Bdfs is especially apparent for the most Bdf-dependent genes where transcription defects significantly exceed the defect in TFIIB binding (*Figure 5C*). Combined, our results suggest an additional role of Bdfs in transcription that extend beyond organizing a platform for recruitment of PIC components.

## Bdf1/2 promote elongation at a subset of genes

At many metazoan promoters, Pol II pauses after transcribing ~20–100 nucleotides, and release of paused Pol II, mediated by phosphorylation of NELF, DSIF, and the C-terminal repeat domain (CTD) Ser2, is a critical step in gene regulation (*Core and Adelman, 2019*). Interestingly, Brd4 was shown to be involved in Ser2 phosphorylation (*Muhar et al., 2018*; *Winter et al., 2017*). While metazoan-like Pol II pausing does not occur at yeast promoters, Pol II stalling and shifts in Pol II distribution under specific conditions have been observed in both budding and fission yeast (*Badjatia et al., 2021*; *Shetty et al., 2017*).

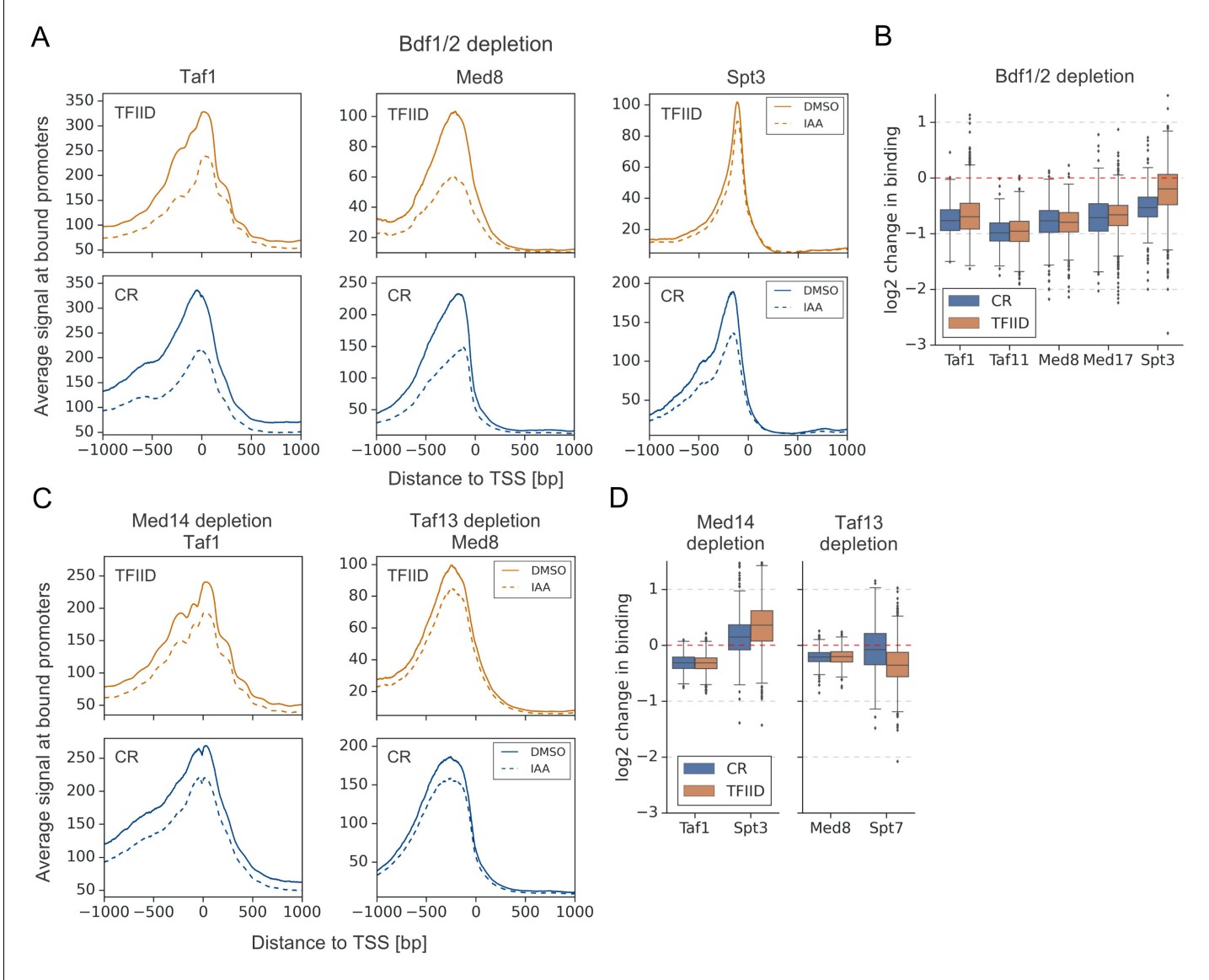

**Figure 4.** Bdf1/2 participate in recruitment of TFIID and Mediator to chromatin. (**A**) Average plots comparing Taf1, Med8, and Spt3 ChEC-seq signals before (dimethyl sulfoxide [DMSO], solid line) and after (indole-3-acetic acid [IAA], dashed line) Bdf1/2 depletion at promoters bound by each factor and classified into TFIID-dependent and coactivator-redundant (CR) categories (2879, 890, and 2526 promoters, respectively). Mean values from replicate experiments are plotted. (**B**) Boxplot showing $\log_2$ change in promoter occupancy of indicated factors after Bdf1/2 depletion. Signals were calculated in a 200 bp window centered on a dominant peak. (**C**) Average plots comparing Taf1 and Med8 ChEC-seq signals before (DMSO, solid line) and after (IAA, dashed line) Med14 or Taf13 depletion, respectively. Same set of promoters as in (**A**) was used for this analysis. (**D**) Boxplot showing $\log_2$ change in promoter occupancy of indicated factors after Med14 or Taf13 depletion. Signals were calculated in a 200 bp window centered on a dominant peak. List of Spt3 bound promoters was used to calculate Spt7 occupancy. See also *Supplementary file 3*.

The online version of this article includes the following figure supplement(s) for figure 4:

**Figure supplement 1.** Bdf1/2 participate in recruitment of TFIID and Mediator to chromatin.

We tested whether Bdfs, like the mammalian BET factors, play a role in Pol II elongation. Pol II was quantitated using ChIP-seq for Rpb1 before and after depleting Bdfs and, as a comparison, Taf1. We first calculated the change in Pol II occupancy along the whole transcribed region. We found that the change in transcription following Bdf depletion is not proportional to the loss of Pol II signal, even though the two results show a good correlation (*Figure 6A* and *Supplementary file 4*). As we observed for TFIIB, the decrease in Pol II signal at the most Bdf-dependent genes is less severe than the loss of transcription. Importantly, we did not detect such difference after depleting

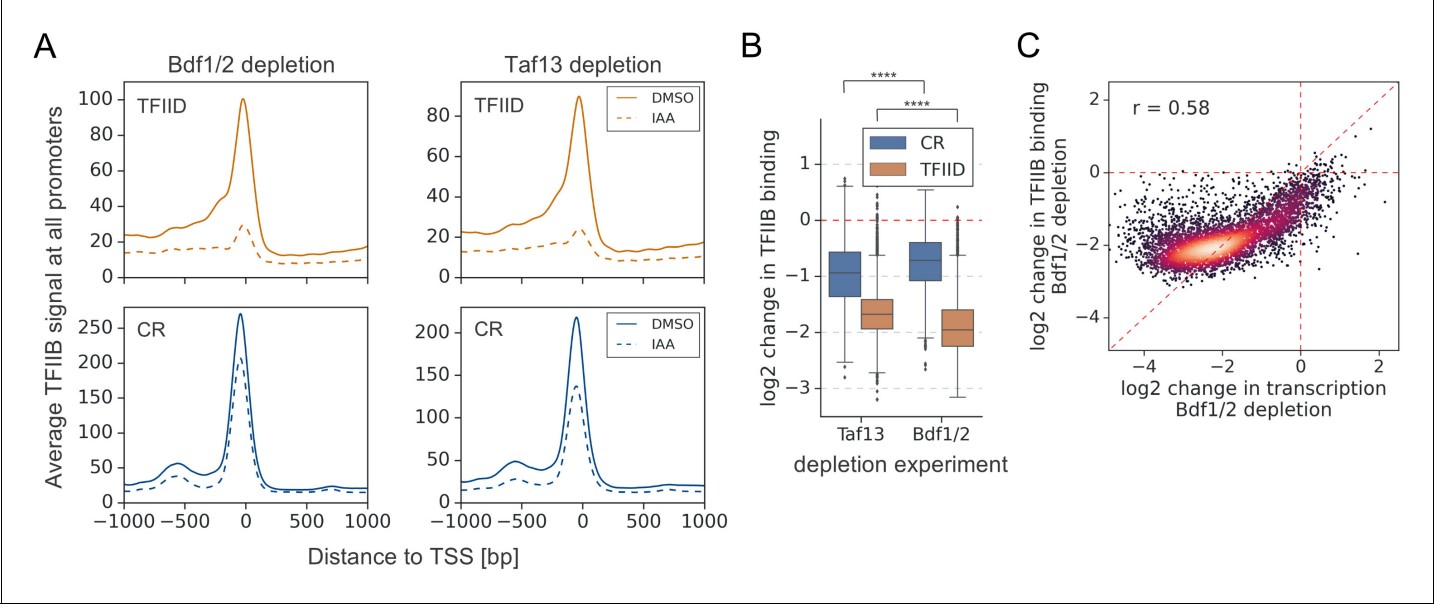

**Figure 5.** Depletion of Bdf1/2 compromises TFIIB recruitment to promoters. (**A**) Average plots comparing TFIIB ChIP-seq signals before (dimethyl sulfoxide [DMSO], solid line) and after (indole-3-acetic acid [IAA], dashed line) Bdf1/2 or Taf13 degradation at 4900 promoters classified into TFIID-dependent and coactivator-redundant (CR) categories. Mean values from replicate experiments are plotted. (**B**) Boxplot showing $\log_2$ change in promoter occupancy of TFIIB after Taf13 or Bdf1/2 degradation. Signals were calculated in a −200 to 100 bp window relative to transcription start site (TSS). Results of the Welch's t-test are shown. (**C**) Scatter plot comparing $\log_2$ change in transcription and $\log_2$ change in TFIIB occupancy after Bdf1/2 degradation at 4883 promoters/genes analyzed in RNA-seq experiments. Spearman correlation coefficient (*r*) is shown. See also *Supplementary file 3*. The online version of this article includes the following figure supplement(s) for figure 5:

**Figure supplement 1.** Genome browser images showing TFIIB ChIP-seq signals at a representative genomic location before (dimethyl sulfoxide [DMSO]) and after (indole-3-acetic acid [IAA]) Bdf1/2 or Taf13 degradation.

Taf1 (*Figure 6—figure supplement 1A* and *Supplementary file 4*). Next, we divided genes into quintiles based on Bdf dependence and plotted the change in Pol II occupancy up to 1 kb downstream from the transcription start site (TSS) (*Figure 6B*). Except for the first quintile (the least Bdf-dependent genes), Bdf depletion caused a similar loss of Pol II at TSSs of all genes. Interestingly, the Pol II loss increased further downstream of TSS at the most Bdf-dependent genes until stabilizing at ~400 bp downstream of TSS. Conversely, at the least Bdf-dependent genes, the Pol II signal partially recovered in the first ~400 bp downstream of TSS relative to the loss at TSS. We calculated the Pol II traveling ratio (TR) defined as the ratio of Rpb1 signal at 5′ versus the 3′ end of transcribed region (*Rahl et al., 2010*). We observed a decrease of TR at the least Bdf-dependent genes and an increase of TR at the most dependent genes following Bdf depletion (*Figure 6C* and *Supplementary file 4*). Analysis of individual genes revealed shifts in Pol II distribution towards 3′ or 5′ ends of genes, respectively (*Figure 6—figure supplement 1B*). We did not detect similar changes in Pol II distribution after depleting Taf1 (*Figure 6—figure supplement 2A, B* and *Supplementary file 4*). Importantly, the change in TR after depleting Bdfs correlates with the change in transcription at individual genes (*Figure 6D*). There is also a modest correlation between the change in TR and the loss of TFIIB at promoters (*Figure 6—figure supplement 2C*). Combined with our earlier findings, this suggests that at least a subset of the most Bdf-dependent genes experience transcription defects both at the initiation and elongation stages upon Bdf depletion. This explains the disproportionally large decrease in the amount of nascent RNA. On the other hand, more efficient elongation at the least Bdf-dependent genes seems to partially nullify the initiation defects, leading to little total change in transcription upon Bdf depletion (*Figure 1C*).

To further explore elongation defects, we performed ChIP-seq with antibodies specific to phosphorylated CTD residues Ser5P and Ser2P. Sequential deposition of these marks coordinates the assembly of multiple factors during transcription, and they are involved in successful promoter escape and elongation, respectively (*Harlen and Churchman, 2017*). We measured total Ser5P and

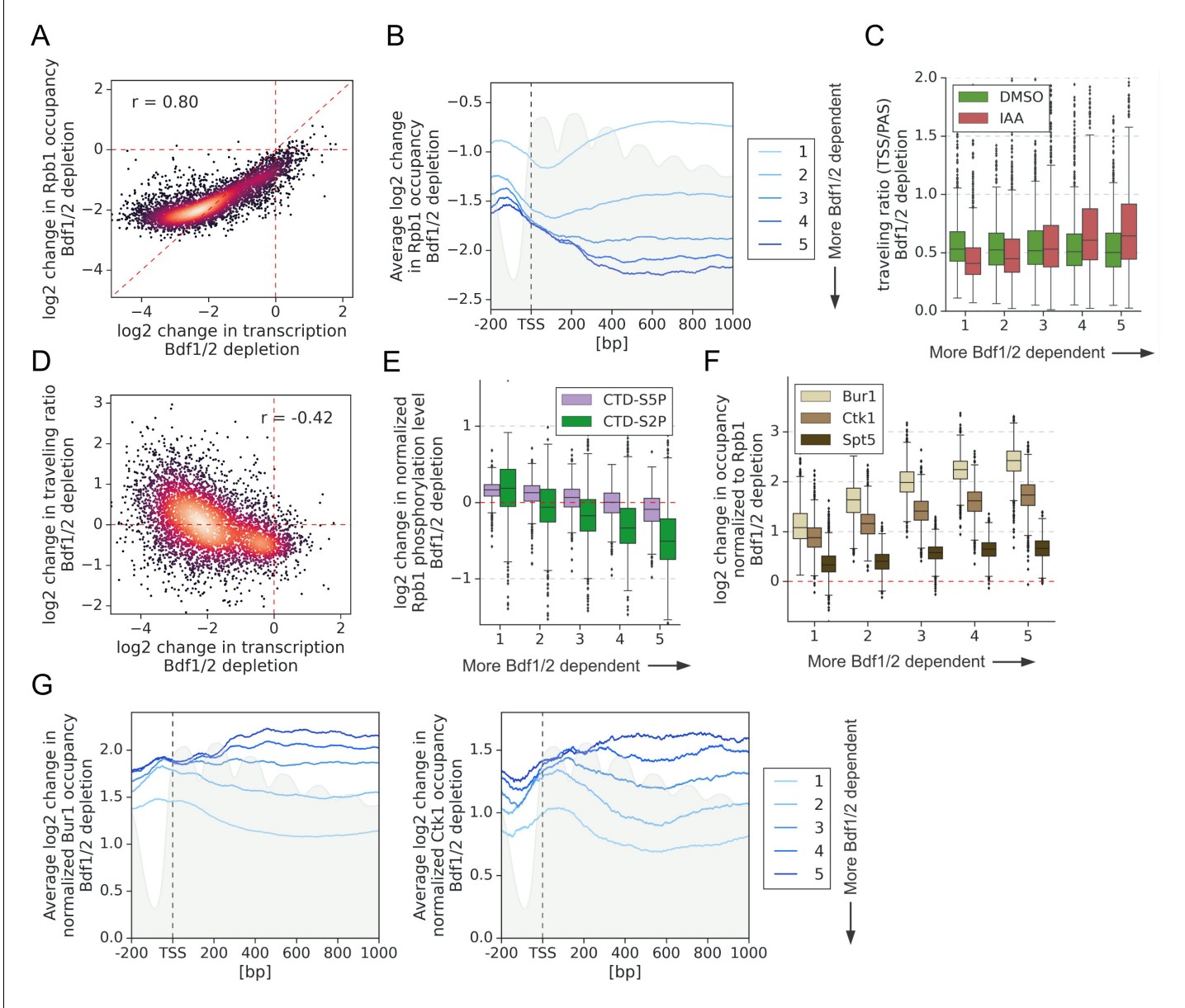

**Figure 6.** Bdf1/2 are involved in transcription elongation. (**A**) Scatter plot comparing log$_2$ change in transcription and occupancy of the largest Pol II subunit Rpb1 after Bdf1/2 degradation. Rpb1 occupancy was calculated along the whole transcribed region for 4615 genes longer than 300 bp, and with annotated transcription start site (TSS) and polyadenylation site (PAS) locations (**Park et al., 2014**). Spearman correlation coefficient (*r*) is shown. Mean values from replicate experiments are plotted. (**B**) Average plot showing log$_2$ change in Rpb1 occupancy from 200 bp upstream to 1000 bp downstream of TSS after depleting Bdf1/2. Data in this and following panels are divided into five groups based on gene dependence on Bdf1/2 as measured by 4-thioU RNA-seq. Data for 3438 genes longer than 1 kb and with annotated TSS and PAS locations (**Park et al., 2014**) are plotted. Published MNase-seq dataset is shown as a gray area plot (**Oberbeckmann et al., 2019**). (**C**) Boxplot comparing Pol II traveling ratio (TR) in dimethyl sulfoxide (DMSO) or indole-3-acetic acid (IAA)-treated samples in the Bdf1/2 degron experiment. TR represents a ratio of Rpb1 occupancy calculated in 100 bp windows at the beginning and end of a transcribed region. The same set of 4615 genes as in (**A**) was used. (**D**) Scatter plot comparing log$_2$ change in transcription and log$_2$ change in Pol II TR after depleting Bdf1/2. Spearman correlation coefficient (*r*) is shown. (**E**) Boxplot comparing log$_2$ change in Rpb1 CTD phosphorylation status at Ser2 and Ser5 residues after depleting Bdf1/2. Data were calculated along the whole transcribed region for the same set of 4615 genes as in (**A**) and signals were normalized to the total Rpb1 signal. (**F**) Boxplot comparing log$_2$ change in Bur1, Ctk1, and Spt5 occupancy after depleting Bdf1/2. Data for DMSO and IAA-treated samples were calculated along the whole transcribed region for the same set of 4615 genes as in (**A**) and signals were normalized to the total Rpb1 signal. See also **Supplementary file 4**. (**G**) Average plots showing log$_2$ changes in Bur1 and Ctk1 occupancy from 200 bp upstream to 1000 bp downstream of TSS after depleting Bdf1/2. Data for 3438 genes longer than 1 kb and with annotated TSS and PAS locations (**Park et al., 2014**) are plotted. Published MNase-seq dataset is shown as a gray area plot (**Oberbeckmann et al., 2019**).

*Figure 6 continued on next page*

*Figure 6 continued*

The online version of this article includes the following source data and figure supplement(s) for figure 6:

**Source data 1.** Processed data used to prepare average line plot in (B) showing log$_2$ change in Rpb1 occupancy following Bdf1/2 depletion.
**Source data 2.** Processed data used to prepare average line plot in (G) showing log$_2$ change in Bur1 occupancy following Taf1 depletion.
**Source data 3.** Processed data used to prepare average line plot in (G) showing log$_2$ change in Ctk1 occupancy following Taf1 depletion.
**Figure supplement 1.** Bdf1/2 are involved in transcription elongation.
**Figure supplement 2.** TFIID depletion does not affect transcription elongation.
**Figure supplement 2—source data 1.** Processed data used to prepare average line plot in (A) showing log$_2$ change in Rpb1 occupancy following Taf1 depletion.
**Figure supplement 3.** Average plots showing Bur1, Ctk1, and Spt5 occupancy from 200 bp upstream to 1000 bp downstream of transcription start site (TSS).
**Figure supplement 3—source data 1.** Processed data used to prepare average line plot showing Bur1 occupancy.
**Figure supplement 3—source data 2.** Processed data used to prepare average line plot showing Ctk1 occupancy.
**Figure supplement 3—source data 3.** Processed data used to prepare average line plot showing Spt5 occupancy.

Ser2P signals along transcribed regions, normalized them by corresponding Rpb1 signals, and calculated the change in CTD phosphorylation status following Bdf depletion (*Figure 6E* and *Supplementary file 4*). In agreement with the above findings that the most Bdf-dependent genes have a defect in elongation, Pol II becomes hypophosphorylated on Ser2 and, to a lesser extent, on Ser5 after depleting Bdfs. Conversely, at the least Bdf-dependent genes, Pol II becomes weakly hyperphosphorylated, in agreement with the suggestion that elongation becomes more efficient at this gene set upon Bdf depletion. Importantly, we did not observe similar changes in Pol II phosphorylation after Taf1 depletion, which validates that Bdfs have a TFIID-independent role in regulating transcriptional elongation (*Figure 6—figure supplement 2D* and *Supplementary file 4*).

## CTD kinases Bur1 and Ctk1 accumulate at the most Bdf1/2-dependent genes following Bdf1/2 depletion

Bur1 and Ctk1, homologues of metazoan Cdk9 and Cdk12/13, are responsible for CTD Ser2P marks in yeast. We used ChIP-seq to measure Bur1 and Ctk1 levels along the transcribed regions and normalized them by corresponding Rpb1 signals. As a comparison, we did a similar analysis for the elongation factor Spt5. Interestingly, the relative gene body occupancies of Bur1 and Ctk1, but not of Spt5, correlate with gene dependence on Bdfs (*Figure 6—figure supplement 3*). Next, we calculated the change in Bur1 and Ctk1 relative occupancy following Bdf depletion. We observed approximately twofold increases in occupancy of both kinases relative to Rpb1 at the least Bdf-dependent genes, a result that agrees with observed CTD hyperphosphorylation. Surprisingly, at the rest of the genes, the relative occupancy of both kinases further increased, correlating with gene dependence on Bdfs. In contrast, we observed a uniform, small (~1.4-fold) increase in Spt5 occupancy, normalized to Rpb1, at all tested genes (*Figure 6F* and *Supplementary file 4*). We divided genes into quintiles based on Bdf dependence and plotted the change in Bur1 and Ctk1 occupancy up to 1 kb downstream from TSS (*Figure 6G*). Similarly as for Rpb1, but showing the opposite trend, Bur1 and Ctk1 accumulate at the 5' end of the most Bdf-dependent genes until stabilizing at ~400 bp downstream of TSS. These results illustrate that changes in the recruitment of Spt5 and CTD kinases cannot account for the function of Bdfs in transcription elongation. They also provide further evidence that the observed defect in elongation is restricted to the first ~400 bp in the transcribed region.

## Discussion

After development of specific inhibitors targeting BET BDs, BET factors, especially Brd4, emerged as key regulators of transcription and as promising targets in the therapy of cancer and immunoinflammatory diseases (*Fujisawa and Filippakopoulos, 2017*; *Wang et al., 2021*). In this work, we investigated the roles of yeast BET family members Bdf1 and Bdf2. We explored transcription dependence on Bdf1/2, their genome-wide binding patterns, and their interplay with key components of the transcriptional machinery. We found that Bdf functions extend beyond solely regulating TFIID as expression from most genes is more sensitive to Bdf versus TFIID depletion. Our results

establish the Bdfs as critical for normal expression of most yeast genes, with functions both during transcription initiation and elongation. Our work reveals that the roles of yeast BET factors are surprisingly similar to their human equivalents and suggests conserved mechanisms of BET function across eukaryotes.

Bdf1 was proposed to act as part of yeast TFIID, substituting for the metazoan Taf1 BDs (*Matangkasombut et al., 2000*). This function might be especially important in yeast as H4 acetylation is enriched at most promoters and yeast promoters lack known DNA sequence motifs that are bound by TFIID. However, we found that that the organization and amino acid sequence of Bdf1 is more similar to human BET factors than to the hTaf1 BDs. Our results showed that Bdfs regulate a similar set of genes as TFIID, but that there are important differences. Transcription from most genes defined earlier as CR is largely insensitive to Bdf depletion, while a large subset of TFIID-dependent genes is significantly more affected by Bdf versus TFIID depletion.

Genome-wide mapping revealed that Bdf1/2 are found together with TFIID at many promoters and that they contribute to TFIID recruitment. However, a substantial fraction of TFIID is still bound to promoters in the absence of Bdfs, in agreement with the sub-stoichiometric association between Bdf1 and TFIID (*Sanders et al., 2002*). Interestingly, human BET proteins were shown to associate with TFIID in a proteomic screen and both Brd4 and hTaf1 were proposed to have synergistic effects on gene expression and cell growth (*Lambert et al., 2019*; *Sdelci et al., 2016*). Finally, hTaf1 BDs have the same substrate specificity as BET BDs and both Brd4 and TFIID are bound to the majority of active promoters in human cells (*Bhagwat et al., 2016*; *Lauberth et al., 2013*; *Slaughter et al., 2021*). Based on our findings and the presented evidence from human cells, we propose that cooperation between BET factors and TFIID is a conserved feature of eukaryotic gene regulation. The gain of BDs by metazoan TFIID may result from the increased complexity of gene expression programs and allow for BET-independent TFIID interactions with acetylated histone H4.

Brd4 interacts with Mediator and participates in Mediator recruitment to enhancers and promoters (*Bhagwat et al., 2016*; *Wu et al., 2003*). Our experiments revealed that Bdfs are important for normal levels of Mediator occupancy. A direct role of Bdfs in Mediator targeting to chromatin was surprising, and we considered the possibility that other factors are involved in that process. Mediator was proposed to be recruited cooperatively with TFIID (*Grünberg et al., 2016*; *Johnson et al., 2002*; *Knoll et al., 2018*). We confirmed this modest cooperativity, but we found that the contribution of Bdfs to Mediator recruitment surpasses that of TFIID. There is strong evidence that TFIID and Mediator act as molecular scaffolds to organize the recruitment of other components of transcriptional machinery (*Allen and Taatjes, 2015*; *Patel et al., 2020*). Human Mediator and Brd4 were also found to participate in the formation of dynamic nuclear condensates at the sites of active transcription (*Han et al., 2020*; *Sabari et al., 2018*). Finally, a recent study suggested cooperation of yeast TFIID and Mediator in restricting the diffusion of PIC components to shared subnuclear territories in order to facilitate gene transcription (*Nguyen et al., 2020*). We propose that chromatin tethered yeast BET factors serve as a nucleation center for dynamic recruitment of Mediator and TFIID, which in turn create a platform for efficient PIC assembly. This role of Bdf1/2 seems widespread and likely complementary to the action of sequence-specific TFs (*Figure 7*).

At the least Bdf-dependent genes, we observed a modest decrease in TFIID, Mediator, and TFIIB recruitment after Bdf depletion, which did not translate into appreciable defects in mRNA synthesis. Conversely, at the most Bdf-dependent genes, the loss of transcription exceeded the loss of Mediator, TFIID, TFIIB, and Pol II recruitment. The unexpected role of the yeast BET factors in transcription elongation provided an explanation for these seemingly conflicting results (*Figure 7*). At CR genes and at a small subset of TFIID-dependent genes, Bdf depletion results in a 3′ shift in Pol II distribution and increased CTD Ser2 phosphorylation. This suggests more efficient early elongation of these genes in the absence of Bdfs. In contrast, Bdf depletion at many TFIID-dependent genes causes Pol II accumulation at 5′ gene ends and a loss of CTD Ser2 phosphorylation. This change in Pol II profile is similar to the defects in pause release caused by BET depletion in human cells (*Muhar et al., 2018*; *Winter et al., 2017*). At these yeast genes, the combined initiation and elongation defects resulting from Bdf depletion lead to a large decrease in mRNA synthesis. Interestingly, it was recently shown that acute stress in *Saccharomyces cerevisiae* cells causes Pol II stalling at the +2 nucleosome, which overlaps with the region where we observe dominant changes in Pol II distribution (*Badjatia et al., 2021*).

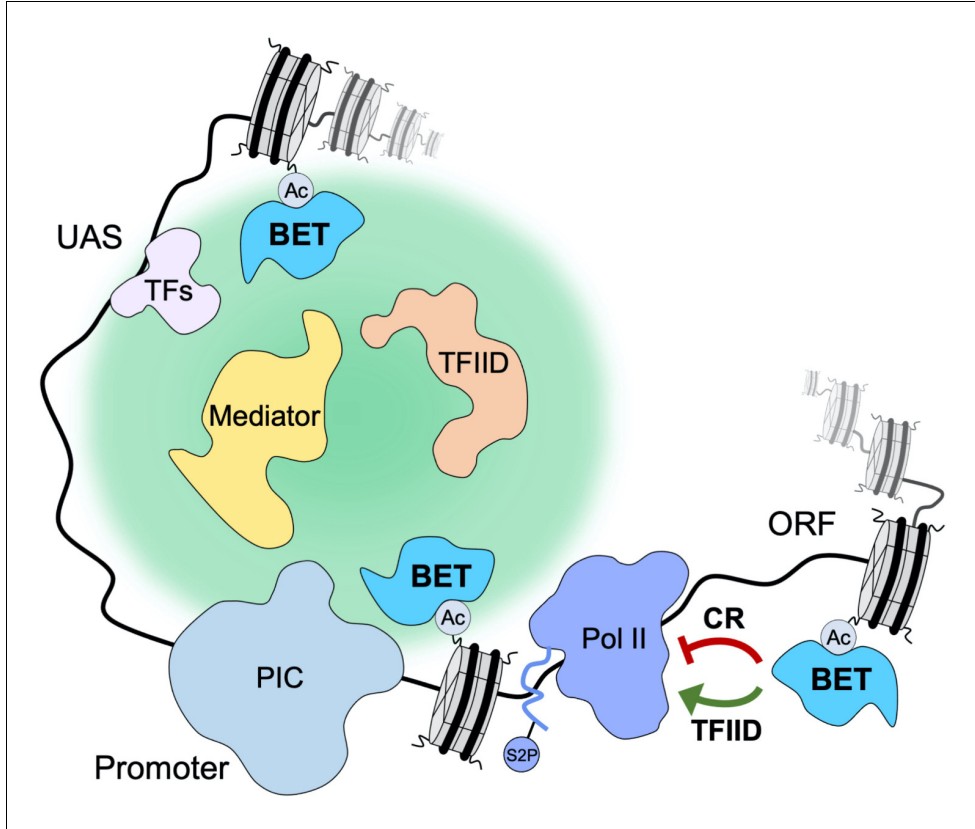

**Figure 7.** Yeast bromodomain and extra-terminal domain (BET) proteins regulate transcription initiation and elongation. Yeast BET proteins bind to regions of the genome highly acetylated at histone H4. Together with sequence-specific transcription factors (TFs), they provide a nucleation center for dynamic recruitment of TFIID and Mediator to the promoter-proximal nuclear territory. In turn, TFIID and Mediator create a platform that facilitates preinitiation complex (PIC) assembly and stimulates transcription initiation. Yeast BET proteins also affect early elongation serving as negative regulatory factors at coactivator-redundant (CR) genes and positive regulatory factors at TFIID-dependent genes.

CTD Ser2 residues in yeast are phosphorylated by kinases Bur1 and Ctk1, homologues of metazoan Cdk9 and Cdk12/13, respectively. Ctk1 is the dominant CTD kinase in yeast while in metazoans Cdk12/13 seems to be less important, although the interplay between Cdk9 and Cdk12/13 is poorly understood (*Harlen and Churchman, 2017*). At Bdf-independent genes, depletion of Bdfs results in a modest increase of Bur1 and Ctk1 occupancy normalized to Rpb1 level, and this agrees with a gain of CTD Ser2P marks at these genes. Surprisingly, at Bdf-dependent genes, the occupancy of both kinases relative to Rpb1 increased significantly in the first ~400 bp of transcribed region after Bdf depletion, even though CTD Ser2 phosphorylation decreases. Interestingly, depletion of Brd4 or all BET factors in human cells was shown to result in a small increase in Cdk9 and Cyclin T1 chromatin occupancy, even though it causes a decrease in CTD Ser2P marks and a defect in pause release (*Muhar et al., 2018*; *Winter et al., 2017*). It is not known if Brd4 affects Cdk12/13 recruitment. Several models have been proposed to explain the role of Brd4 in transcription elongation. Brd4 was shown to release Hexim1-mediated inhibition of Cdk9 in vitro, thus acting as a positive regulator of P-TEFb (*Itzen et al., 2014*). It was also proposed that Brd2 and Brd4 have kinase activity and that Brd4 is directly connected to CTD Ser2 phosphorylation independently of P-TEFb (*Denis and Green, 1996*; *Devaiah et al., 2012*). Bdf1 was also proposed to have an intrinsic kinase activity, which raises an intriguing prospect of its direct involvement in CTD phosphorylation that, if correct, would provide an explanation for our results (*Matangkasombut et al., 2000*). Finally, it is possible that CTD hypophosphorylation at BET-dependent genes may result from defects in recruitment and/or reduction in function of other elongation factors. Since the role(s) of the BET factors in elongation are

unresolved in both systems, it will be of great interest to determine whether the yeast and metazoan factors use similar mechanisms to regulate elongation.

Redundancy between three (Brd2, Brd3, Brd4) of the four mammalian BET factors was proposed based on their ubiquitous expression in most tissues, similar binding patterns, and because they are all essential for embryonic development (*Gyuris et al., 2009*; *Khoueiry et al., 2019*; *Uhlén et al., 2015*). The extent of this redundancy is poorly understood because only a few context-specific examples of shared functions of BET factors have been reported (*Gilan et al., 2020*; *Rahman et al., 2011*; *Stonestrom et al., 2015*). In most studies, the roles of BET factors in transcriptional regulation were attributed to Brd4 alone (*Muhar et al., 2018*; *Zheng et al., 2021*; *Zuber et al., 2011*). Our data indicate that yeast BET proteins are functionally redundant under normal growth conditions, with Bdf1 being the dominant factor. The changes in transcription following depletion of Bdf1 or Bdf1/2 are highly correlated, while depletion of Bdf2 does not have significant consequences for the cell. In addition, Bdf1 and Bdf2 have nearly identical genome-wide binding patterns and Bdf2 redistributes from CR to TFIID-dependent genes upon Bdf1 depletion. Interestingly, it was reported that Bdf2 overexpression can suppress temperature- and salt-sensitive phenotypes in a *bdf1* deletion strain, but, at the same time, Bdf1 was found to be a negative regulator of Bdf2 expression (*Fu et al., 2013*; *Matangkasombut et al., 2000*). This latter finding is also supported by our RNA-seq results where we observed approximately two-fold upregulation of *BDF2* transcription following Bdf1 depletion. Importantly, a similar interplay between human BET factors in regulating expression of other family members was recently proposed (*Lambert et al., 2019*). It remains to be investigated if and to what extent Brd2 and/or Brd3 can replace Brd4 in supporting transcription in human cells.

Recent years have brought remarkable discoveries highlighting the essential nature of mammalian BET factors. Still, many aspects of their biology remain poorly understood, including details of the relationship with other components of the transcriptional machinery at enhancers and promoters and the extent of redundancies. We uncovered striking similarities between yeast and mammalian BET family members. Many of the most BET-sensitive genes in both systems lack a TATA element in their promoters and control highly similar cellular processes. BET proteins are also universally involved in the recruitment of Mediator to chromatin and modulation of transcriptional elongation. Considering the relative simplicity compared to metazoans, the yeast model system offers unique opportunities for a detailed investigation of different aspects of BET biology, and this work provides a foundation for these future studies.

# Materials and methods

## Key resources table

| Reagent type (species) or resource | Designation | Source or reference | Identifiers | Additional information |
|---|---|---|---|---|
| Strain, strain background (*Saccharomyces cerevisiae*) | BY4705 | PMID:9483801 | ATCC: 200869 | Laboratory of Dan Gottschling; WT |
| Genetic reagent (*Schizosaccharomyces pombe*) | SHY1058 | PMID:28918900 | *h- RPB3-3x Flag::NatMX* | WT |
| Genetic reagent (*S. pombe*) | SHY1110 | PMID:18216783 | *h + leu1-32 ade6-216 OtrR1:ura4 Abp1-3xFlag::KanMX Cbh1-13xMyc::NatMX* | Laboratory of Toshio Tsukiyama; WT |
| Genetic reagent (*S. cerevisiae*) | SHY1036 | PMID:28918900 | *RPB3-3x Flag::NatMX, pGPD1-OSTIR::HIS3* | WT-degron tagging (BY4705 derivative) |
| Genetic reagent (*S. cerevisiae*) | SHY1235 | Laboratory of Steven Hahn | *pGPD1-OSTIR::HIS3* | WT-degron tagging (BY4705 derivative) |
| Genetic reagent (*S. cerevisiae*) | SHY1039 | PMID:31913117 | *RPB3-3x Flag::NatMX, pGPD1-OSTIR::HIS3, TAF1-3xV5 IAA7::KanMX* | Taf1 degron |

*Continued on next page*

*Continued*

| Reagent type (species) or resource | Designation | Source or reference | Identifiers | Additional information |
|---|---|---|---|---|
| Genetic reagent (*S. cerevisiae*) | SHY1059 | Laboratory of Steven Hahn | *RPB3-3x Flag::NatMX, pGPD1-OSTIR::HIS3, bdf2△::HPH* | *bdf2△* |
| Genetic reagent (*S. cerevisiae*) | RDY27 | Laboratory of Steven Hahn | *RPB3-3x Flag::NatMX, pGPD1-OSTIR::HIS3, BDF1-3xV5 IAA7::KanMX* | Bdf1 degron |
| Genetic reagent (*S. cerevisiae*) | RDY28 | Laboratory of Steven Hahn | *RPB3-3x Flag::NatMX, pGPD1-OSTIR::HIS3, BDF2-3xV5 IAA7::KanMX* | Bdf2 degron |
| Genetic reagent (*S. cerevisiae*) | RDY70 | Laboratory of Steven Hahn | *pGPD1-OSTIR::HIS3, BDF2-3xV5 IAA7::KanMX* | Bdf2 degron |
| Genetic reagent (*S. cerevisiae*) | RDY73 | Laboratory of Steven Hahn | *pGPD1-OSTIR::HIS3, BDF2-3xV5 IAA7::KanMX, BDF1-3xV5 IAA7::URA3* | Bdf1/2 degron |
| Genetic reagent (*S. cerevisiae*) | RDY1 | Laboratory of Steven Hahn | *RPB3-3x Flag::NatMX, pGPD1-OSTIR::HIS3, bdf2△::HPH, BDF1-3xV5 IAA7::KanMX* | *bdf2△*, Bdf1 degron |
| Genetic reagent (*S. cerevisiae*) | SHY1137 | Laboratory of Steven Hahn | *RPB3-3x Flag::NatMX, pGPD1-OSTIR::HIS3, TOA1-3xV5 IAA7* | Toa1 degron |
| Genetic reagent (*S. cerevisiae*) | SHY1213 | Laboratory of Steven Hahn | *pGPD1-OSTIR::HIS3, ssl2△::KanMX/pSH1947 (LEU2 SSL2-3xV5 IAA7)* | Ssl2 degron |
| Genetic reagent (*S. cerevisiae*) | RDY29 | Laboratory of Steven Hahn | *RPB3-3x Flag::NatMX, pGPD1-OSTIR::HIS3, HTZ1-3xV5 IAA7::KanMX* | H2A.Z degron |
| Genetic reagent (*S. cerevisiae*) | RDY57 | Laboratory of Steven Hahn | *RPB3-3x Flag::NatMX, pGPD1-OSTIR::HIS3, esa1△::KanMX/ pRD8 (LEU2 ESA1-3xV5 IAA7)* | Esa1 degron |
| Genetic reagent (*S. cerevisiae*) | RDY44 | Laboratory of Steven Hahn | *pBDF2-MNase::TRP1* | Free MNase (*BDF2* promoter) |
| Genetic reagent (*S. cerevisiae*) | RDY34 | Laboratory of Steven Hahn | *RPB3-3x Flag::NatMX, pGPD1-OSTIR::HIS3, BDF1-3xV5 IAA7, BDF2-MNase::TRP1* | Bdf1 degron, Bdf2-MNase |
| Genetic reagent (*S. cerevisiae*) | RDY33 | Laboratory of Steven Hahn | *RPB3-3x Flag::NatMX, pGPD1-OSTIR::HIS3, BDF2-3xV5 IAA7, BDF1-MNase::TRP1* | Bdf2 degron, Bdf1-MNase |
| Genetic reagent (*S. cerevisiae*) | RDY59 | Laboratory of Steven Hahn | *RPB3-3x Flag::NatMX, pGPD1-OSTIR::HIS3, esa1△::KanMX, BDF1-MNase::TRP1/pRD8 (LEU2 ESA1-3xV5 IAA7)* | Esa1 degron, Bdf1-MNase |
| Genetic reagent (*S. cerevisiae*) | RDY85 | Laboratory of Steven Hahn | *pGPD1-OSTIR::HIS3, BDF1-3xFlag::NatMX* | Bdf1-3xFlag |
| Genetic reagent (*S. cerevisiae*) | RDY86 | Laboratory of Steven Hahn | *pGPD1-OSTIR::HIS3, BDF2-3xFlag::NatMX* | Bdf2-3xFlag |
| Genetic reagent (*S. cerevisiae*) | RDY30 | Laboratory of Steven Hahn | *RPB3-3x Flag::NatMX, pGPD1-OSTIR::HIS3, bdf2△::HygMX, BDF1-3xV5 IAA7::KanMX, TAF1-MNase::TRP1* | *bdf2△*, Bdf1 degron, Taf1-MNase |

*Continued on next page*

*Continued*

| Reagent type (species) or resource | Designation | Source or reference | Identifiers | Additional information |
|---|---|---|---|---|
| Genetic reagent (*S. cerevisiae*) | RDY21 | Laboratory of Steven Hahn | *RPB3-3x Flag::NatMX, pGPD1-OSTIR::HIS3, bdf2△::HygMX, BDF1-3xV5 IAA7::KanMX, TAF11-MNase::TRP1* | *bdf2△*, Bdf1 degron, Taf11-MNase |
| Genetic reagent (*S. cerevisiae*) | RDY26 | Laboratory of Steven Hahn | *RPB3-3x Flag::NatMX, pGPD1-OSTIR::HIS3, bdf2△::HygMX, BDF1-3xV5 IAA7::KanMX, MED8-MNase::TRP1* | *bdf2△*, Bdf1 degron, Med8-MNase |
| Genetic reagent (*S. cerevisiae*) | RDY50 | Laboratory of Steven Hahn | *RPB3-3x Flag::NatMX, pGPD1-OSTIR::HIS3, bdf2△::HygMX, BDF1-3xV5 IAA7::KanMX, MED17-MNase::TRP1* | *bdf2△*, Bdf1 degron, Med17-MNase |
| Genetic reagent (*S. cerevisiae*) | RDY51 | Laboratory of Steven Hahn | *RPB3-3x Flag::NatMX, pGPD1-OSTIR::HIS3, bdf2△::HygMX, BDF1-3xV5 IAA7::KanMX, SPT3-MNase::TRP1* | *bdf2△*, Bdf1 degron, Spt3-MNase |
| Genetic reagent (*S. cerevisiae*) | SHY1172 | Laboratory of Steven Hahn | *pGPD1-OSTIR::HIS3, MED14-3xV5 IAA7::KanMX, SPT3-MNase::TRP1* | Med14 degron, Spt3-MNase |
| Genetic reagent (*S. cerevisiae*) | SHY1301 | Laboratory of Steven Hahn | *pGPD1-OSTIR::HIS3, MED14-3xV5 IAA7::KanMX, TAF1-MNase::TRP1* | Med14 degron, Taf1-MNase |
| Genetic reagent (*S. cerevisiae*) | SHY1327 | Laboratory of Steven Hahn | *pGPD1-OSTIR::HIS3, TAF13-3xV5 IAA7::KanMX, MED8-MNase::TRP1* | Taf13 degron, Med8-MNase |
| Genetic reagent (*S. cerevisiae*) | SHY1331 | Laboratory of Steven Hahn | *pGPD1-OSTIR::HIS3, TAF13-3xV5 IAA7::KanMX, SPT7-MNase::TRP1* | Taf13 degron, Spt7-MNase |
| Genetic reagent (*S. cerevisiae*) | RDY82 | Laboratory of Steven Hahn | *pGPD1-OSTIR::HIS3, BDF2-3xV5 IAA7::KanMX, BDF1-3xV5 IAA7::URA3, TAF1-3xFlag::NatMX* | Bdf1/2 degron, Taf1-3xFlag |
| Genetic reagent (*S. cerevisiae*) | SHY1089 | Laboratory of Steven Hahn | *RPB3-3x Flag::NatMX, pGPD1-OSTIR::HIS3, TAF13-3xV5 IAA7::KanMX, SUA7-13xMyc::HPH* | Taf13 degron, TFIIB-13xMyc |
| Genetic reagent (*S. cerevisiae*) | RDY92 | Laboratory of Steven Hahn | *pGPD1-OSTIR::HIS3, BDF2-3xV5 IAA7::KanMX, BDF1-3xV5 IAA7::URA3, SUA7-3xFlag::NatMX* | Bdf1/2 degron, TFIIB-3xFlag |
| Genetic reagent (*S. cerevisiae*) | RDY79 | Laboratory of Steven Hahn | *pGPD1-OSTIR::HIS3, BDF2-3xV5 IAA7::KanMX, BDF1-3xV5 IAA7::URA3, BUR1-3xFlag::NatMX* | Bdf1/2 degron, Bur1-3xFlag |
| Genetic reagent (*S. cerevisiae*) | RDY81 | Laboratory of Steven Hahn | *pGPD1-OSTIR::HIS3, BDF2-3xV5 IAA7::KanMX, BDF1-3xV5 IAA7::URA3, CTK1-3xFlag::NatMX* | Bdf1/2 degron, Ctk1-3xFlag |
| Genetic reagent (*S. cerevisiae*) | RDY94 | Laboratory of Steven Hahn | *pGPD1-OSTIR::HIS3, BDF2-3xV5 IAA7::KanMX, BDF1-3xV5 IAA7::URA3, SPT5-3xFlag::NatMX* | Bdf1/2 degron, Spt5-3xFlag |
| Antibody | Rabbit polyclonal anti-Tfg2 | Laboratory of Steven Hahn | Rabbit #1260K | WB (1:2000) |

*Continued on next page*

Continued

| Reagent type (species) or resource | Designation | Source or reference | Identifiers | Additional information |
|---|---|---|---|---|
| Antibody | Mouse monoclonal anti-V5 | Invitrogen | Cat. #R960-25 | WB (1:1000) |
| Antibody | Rabbit monoclonal anti-H4K12ac | Cell Signaling Technology | Cat. #13944 | WB (1:1000) ChIP-seq (1:100) |
| Antibody | Mouse monoclonal anti-Flag | Sigma-Aldrich | Cat. #F3165 | ChIP-seq (1:100) |
| Antibody | Mouse monoclonal anti-Myc | Cell Signaling Technology | Cat. #2276 | ChIP-seq (1:100) |
| Antibody | Mouse monoclonal anti-Rpb1 CTD | Cell Signaling Technology | Cat. #2629 | ChIP-seq (1:50) |
| Antibody | Rabbit monoclonal anti-Phospho-Rpb1 CTD (Ser2) | Cell Signaling Technology | Cat. #13499 | ChIP-seq (1:50) |
| Antibody | Rabbit monoclonal anti-Phospho-Rpb1 CTD (Ser5) | Cell Signaling Technology | Cat. #13523 | ChIP-seq (1:50) |
| Recombinant DNA reagent | pFA6a-3V5-IAA7-KanMX6 | PMID:30192227 | | Degron tagging vector |
| Recombinant DNA reagent | pSH1855 | Laboratory of Steven Hahn | | Degron tagging vector |
| Recombinant DNA reagent | pGZ110 | PMID:26490019 | | MNase tagging vector |
| Recombinant DNA reagent | p2L-3Flag-NAT | Laboratory of Toshio Tsukiyama | | 3xFlag tagging vector |
| Recombinant DNA reagent | pFA6a-13Myc-Hyg | Laboratory of Toshio Tsukiyama | | 13xMyc tagging vector |
| Recombinant DNA reagent | pRS303 | PMID:2659436 | ATTC: 77138 | |
| Recombinant DNA reagent | pRD15 | Laboratory of Steven Hahn | | pRS303 derivative-free MNase under *BDF2* promoter |
| Recombinant DNA reagent | pRS315 | PMID:2659436 | ATTC: 77144 | |
| Recombinant DNA reagent | pRS316 | PMID:2659436 | ATTC: 77145 | |
| Recombinant DNA reagent | pRD5 | Laboratory of Steven Hahn | | pRS315 derivative-*ESA1* gene with native promoter |
| Recombinant DNA reagent | pRD21 | Laboratory of Steven Hahn | | pRS316 derivative-*ESA1* gene with native promoter and degron tag |
| Commercial assay or kit | RiboPure yeast kit | Thermo Fisher Scientific | Cat. #AM1926 | |
| Commercial assay or kit | MyOne Streptavidin C1 Dynabeads | Thermo Fisher Scientific | Cat. #65002 | |
| Commercial assay or kit | Protein G Dynabeads | Thermo Fisher Scientific | Cat. #10003D | |
| Commercial assay or kit | RNeasy Kit | Qiagen | Cat. #74104 | |
| Commercial assay or kit | Qubit HS RNA assay | Thermo Fisher Scientific | Cat. #Q32852 | |

*Continued on next page*

*Continued*

| Reagent type (species) or resource | Designation | Source or reference | Identifiers | Additional information |
|---|---|---|---|---|
| Commercial assay or kit | Qubit HS DNA assay | Thermo Fisher Scientific | Cat. #Q32851 | |
| Commercial assay or kit | Ovation Universal RNA-seq Library Preparation Kit | Tecan | Cat. #0364-A01 | Including *S. cerevisiae* AnyDeplete reagent |
| Chemical compound, drug | MTSEA biotin-XX | Biotium | Cat. #90066-1 | |
| Chemical compound, drug | 4-thiouracil | Sigma-Aldrich | Cat. #440736 | |
| Chemical compound, drug | Indole-3-acetic acid | Sigma-Aldrich | Cat. #I3750 | |
| Software, algorithm | HTseq | PMID:25260700 | | |
| Software, algorithm | Bowtie 2 | PMID:22388286 | | |
| Software, algorithm | HOMER | PMID:20513432 | | |
| Software, algorithm | Clustal Omega | PMID:21988835 | | |
| Software, algorithm | Jalview v 2 | PMID:19151095 | | |
| Software, algorithm | PANTHER v 16.0 | PMID:33290554 | | |

## Yeast cell growth

All *S. cerevisiae* and *Schizosaccharomyces pombe* strains used in this study are listed in *Supplementary file 5* and Key resources table (*Donczew et al., 2020*; *Baker Brachmann et al., 1998*; *Warfield et al., 2017*; *Chen et al., 2008*). *S. cerevisiae* strains were grown in YPD medium (1% yeast extract, 2% peptone, 2% glucose, 20 µg/ml adenine sulfate) at 30°C with shaking. *S. pombe* strains were grown in YE medium (0.5% yeast extract, 3% glucose) at 30°C with shaking. In experiments involving degron depletion of target proteins, *S. cerevisiae* strains were treated with 500 µM IAA dissolved in DMSO or with DMSO alone for 30 min (or 60 min for Esa1-degron strains) to induce protein degradation followed by protocol-specific steps. In RNA-seq experiments, *S. cerevisiae* and *S. pombe* strains were grown to A600 ~1.0. In ChEC-seq experiments, *S. cerevisiae* strains were grown to A600 ~0.6. In ChIP-seq experiments, *S. cerevisiae* and *S. pombe* strains were grown to A600 ~0.8. The number of biological replicates collected in each experiment is listed in *Supplementary file 6*.

## Strain construction

*S. cerevisiae* strains (*Supplementary file 5*) were constructed using standard methods. Proteins were chromosomally tagged by yeast transformation and homologous recombination of PCR-amplified DNA. Plasmid pFA6a-3V5-IAA7-KanMX6 (*Chan et al., 2018*), or a derivative with the *KanMX* marker replaced by *URA3* (pSH1855), was used as a template for generating the IAA7 degron tags. This C-terminal tag contains three copies of the V5 epitope tag followed by the IAA7 degron. For ChEC-seq experiments, proteins were tagged with 3xFLAG-MNase::TRP1 using pGZ110 (*Zentner et al., 2015*). Plasmids p2L-3Flag-NAT and pFA6a-13Myc-Hyg (a gift from Toshio Tsukiyama, Fred Hutch) were used as templates to generate PCR fragments for tagging proteins with 3xFlag and 13xMyc epitope tags. A strain expressing free MNase under control of the native *BDF2* promoter was constructed the following way. First, the *MED8* promoter in pSG79 (*Grünberg et al., 2016*) was

exchanged with the *BDF2* promoter (containing 500 bp upstream DNA from the *BDF2* start codon). A XhoI/SacI fragment containing the *pBDF2*-MNase fusion DNA was inserted to the yeast integrating vector pRS303 (*Sikorski and Hieter, 1989*). The resulting pRD15 plasmid was linearized with BstEII and integrated into strain BY4705. *BDF2* deletion strain was constructed by replacing the *BDF2* gene with the *HPH* marker amplified from plasmid pAG32 (*Goldstein and McCusker, 1999*). *ESA1*-containing vectors were constructed by inserting a XhoI/SacI fragment containing the *ESA1* gene with its native promoter and terminator into vectors pRS316 (pRD5) and pRS315 (pRD21). A plasmid shuffle strain was constructed containing WT *ESA1* on plasmid pRD5 and a chromosomal deletion of *ESA1* in strain SHY1036. pRD21 was modified by replacing nucleotides 301–354 in the *ESA1* ORF with the 3xV5-IAA7 fragment to create the Esa1-degron. Plasmid shuffling was used to replace pRD5 with pRD21 by counter selection on 5-FOA plates.

## Western blot analysis

1 ml cell culture was collected and pelleted from strains after treatment with IAA or DMSO, washed with 500 µl water, then resuspended in 100 µl yeast whole cell extract buffer (60 mM Tris, 6.8, 10% glycerol, 2% SDS, 5% 2-mercaptoethanol, 0.0025% bromophenol blue). After heating for 5 min at 95°C, samples were centrifuged for 5 min at 21K × g and analyzed by SDS-PAGE and western blot where protein signals were visualized by using the Odyssey CLx scanner and quantified using Odyssey Image Studio software (Li-Cor) by generating a standard curve using a titration from WT extract. Each protein analyzed was normalized to the amount of the TFIIF subunit Tfg2.

## Spot assay for yeast growth

*S. cerevisiae* strains were streaked from glycerol stocks on YPD plates and incubated for 3 days at 30°C. Single colonies from freshly grown plates were used to start overnight cultures for the spot assay. After reaching saturation, cultures were diluted to A600 = 1.0 and 10× serial dilutions were prepared. 10 µl of appropriate dilution was spotted on YPD plates, and the plates were incubated for 2–3 days at 30°C to compare growth rates.

## Isolation of newly synthesized RNA

All steps were done essentially as described with minor modifications (*Donczew et al., 2020*). Two (Bdf1, Bdf2, Δ*bdf2*, Bdf1Δ*bdf2*, H2A.Z) or three (Bdf1/2, Toa1, Ssl2) replicate samples were collected for each experiment. 10 ml *S. cerevisiae* or 20 ml *S. pombe* (strain SHY1058) cultures were labeled with 5 mM 4-thiouracil (4-thioU) (Sigma-Aldrich #440736) for 4 min, the cells were pelleted at 3000 × g for 3 min, flash-frozen in liquid N$_2$, and then stored at −80°C for further use. *S. cerevisiae* and *S. pombe* cells were mixed in an 8:1 ratio and total RNA was extracted using reagents from RiboPure yeast kit (Thermo Fisher Scientific #AM1926) using the following volumes: 480 µl lysis buffer, 48 µl 10% SDS, 480 µl phenol/CHCl$_3$/isoamyl alcohol (25:24:1) per *S. cerevisiae* pellet +50 µl *S. pombe* cell solution (from a single *S. pombe* pellet resuspended in 850 µl lysis buffer). Cells were lysed using 1.25 ml zirconia/silica beads (RPI #9834) in a Mini Beadbeater-96 (BioSpec Products) for 5 min. Lysates were spun for 5 min at 16K × g, then the following volumes were combined in a 5 ml tube: 400 µl supernatant, 1400 µl binding buffer, 940 µl 100% ethanol. Samples were processed through Ambion filter cartridges until all sample was loaded, then washed with 700 µl Wash Solution 1, and twice with 500 µl Wash Solution 2/3. After a final spin to remove residual ethanol, RNA was eluted with 25 µl 95°C preheated Elution Solution. The elution step was repeated, and eluates combined. RNA was then treated with DNaseI using 5 µl DNaseI buffer and 4 µl DNaseI for 30 min at 37°C, then treated with Inactivation Reagent for 5 min at RT. Total RNA samples were stored at −20°C for up to 2 months or at −80°C for longer periods.

RNA was biotinylated using 40 µl (~40 µg) total RNA and 4 µg MTSEA biotin-XX (Biotium #90066-1) in the following reaction: 40 µl total 4-thioU-labeled RNA, 20 mM HEPES, 1 mM EDTA, 4 µg MTSEA biotin-XX (80 µl 50 µg/ml diluted stock) in a 400 µl final volume. Biotinylation reactions occurred for 30 min at RT with rotation and under foil. Unreacted MTS-biotin was removed by phenol/CHCl$_3$/isoamyl alcohol (25:24:1) extraction. RNA was precipitated with isopropanol and resuspended in 100 µl nuclease-free H$_2$O. Biotinylated RNA was purified using 80 µl MyOne Streptavidin C1 Dynabeads (Thermo Fisher Scientific #65002) + 100 µl biotinylated RNA for 15 min at RT with rotation and under foil. Prior to use, MyOne Streptavidin beads were washed in a single batch with

$3 \times 3$ ml H$_2$O, $3 \times 3$ ml High Salt Wash Buffer (100 mM Tris, 7.4, 10 mM EDTA, 1 M NaCl, 0.05% Tween-20), blocked in 4 ml High Salt Wash Buffer containing 40 ng/µl glycogen (Millipore Sigma #10901393001) for 1 hr at RT, then resuspended to the original volume in High Salt Wash Buffer. After incubation with biotinylated RNA, the beads were washed $3 \times 0.8$ ml High Salt Wash Buffer, then eluted into 25 µl streptavidin elution buffer (100 mM DTT, 20 mM HEPES, 7.4, 1 mM EDTA, 100 mM NaCl, 0.05% Tween-20) at RT with shaking, then the elution step was repeated, and eluates were combined for a total of 50 µl. Samples were stored at −20℃ until further processing.

10% input (not biotinylated) RNA (4 µl) was diluted into 50 µl streptavidin elution buffer and processed the same as the biotinylated RNA samples to determine the extent of recovery. 50 µl of each input and biotinylated RNA was adjusted to 100 µl with nuclease-free water and purified on RNeasy columns (Qiagen #74104) using the modified protocol. To each 100 µl sample, 350 µl RLT lysis buffer (supplied by the Qiagen kit and supplemented with 10 µl 1% βME per 1 ml RLT) and 250 µl 100% ethanol were added, mixed well, and applied to columns. Columns were washed with 500 µl RPE wash buffer (supplied by the Qiagen kit and supplemented with 35 µl 1% βME per 500 µl RPE), followed by a final 5 min spin at 21K $\times$ g. RNAs were eluted into 14 µl nuclease-free water, and the RNA concentration was measured using Qubit HS RNA assay (Thermo Fisher Scientific #Q32852). Samples were stored at −20℃. One sample per batch prepared in a single day was tested for enrichment of labeled RNA by RT-qPCR, probing both unlabeled and labeled RNA from at least three transcribed genes as previously described (*Donczew et al., 2020*). The purified 4-thioU labeled RNA contained 2–11% contamination of unlabeled RNA.

## Preparation of RNA-seq libraries for NGS

Newly synthesized RNA isolated via 4-thioU labeling and purification was prepared for sequencing using the Ovation Universal RNA-seq Library Preparation Kit with *S. cerevisiae* AnyDeplete reagent (Tecan #0364-A01) according to the manufacturer's instructions and 50 ng input RNA. Libraries were sequenced on the Illumina HiSeq2500 platform using 25 bp paired-end reads at the Fred Hutchinson Genomics Shared Resources facility.

## ChEC-seq experiments

ChEC-seq was performed as previously described (*Donczew et al., 2021*; *Donczew et al., 2020*). In experiments involving Bdf1, Bdf2, or Bdf1/2 depletion, six replicate samples were collected. In Esa1 depletion experiment, two replicate samples were collected, as well as in experiments mapping SAGA subunits (Spt3, Spt7) following Med14 or Taf13 depletion. In all other experiments, three replicate samples were collected. *S. cerevisiae* 50 ml cultures were pelleted at 2000 $\times$ g for 3 min. Cells were resuspended in 1 ml of Buffer A (15 mM Tris, 7.5, 80 mM KCl, 0.1 mM EGTA, 0.2 mM spermine [Millipore Sigma #S3256], 0.3 mM spermidine [Millipore Sigma #85558], protease inhibitors [Millipore Sigma #04693159001]), transferred to a 1.5 ml tube, and pelleted at 1500 $\times$ g for 30 s. Cell were washed twice with 1 ml of Buffer A and finally resuspended in 570 µl of Buffer A. 30 µl 2% digitonin (Millipore Sigma #300410) was added to a final concentration of 0.1%, and cells were permeabilized for 5 min at 30℃ with shaking (900 rpm). 0.2 mM CaCl$_2$ was added to the samples followed by incubation for another 5 min at 30℃. 100 µl cell suspension was mixed with 100 µl Stop Solution (400 mM NaCl, 20 mM EDTA, 4 mM EGTA). Stop Solution was supplemented with 5 µg MNase digested *Drosophila melanogaster* chromatin. Samples were incubated with 0.4 mg/ml Proteinase K (Thermo Fisher Scientific #AM2548) for 30 min at 55℃, and the DNA was purified by phenol/CHCl$_3$/isoamyl alcohol (25:24:1) extraction and ethanol precipitation. Pellets were resuspended in 30 µl 0.3 mg/ml RNase A (Thermo Fisher Scientific #EN0531) (10 mM Tris, 7.5, 1 mM EDTA, 0.3 mg/ml RNase A) and incubated for 15 min at 37℃. 60 µl of Mag-Bind reagent (Omega Biotek #M1378-01) was added, and the samples were incubated for 10 min at RT. Supernatants were transferred to a new tube, and the volume was adjusted to 200 µl (10 mM Tris, 8.0, 100 mM NaCl). DNA was purified again by phenol/CHCl$_3$/isoamyl alcohol (25:24:1) extraction and ethanol precipitation, and resuspended in 25 µl 10 mM Tris, 8.0.

## ChIP-seq experiments

ChIP-seq experiments were performed similarly as described (*Donczew et al., 2020*). Two replicate samples were collected for all experiments except for Bur1 ChIP-seq, where three replicates were

collected. 100 ml *S. cerevisiae* or *S. pombe* cultures were crosslinked with 1% formaldehyde (Sigma-Aldrich #252549) for 20 min in the above growth conditions, followed by another 5 min treatment with 130 mM glycine. Cells were pelleted at 3000 × g for 5 min, washed with cold TBS buffer, pelleted at 2000 × g for 3 min, flash-frozen in liquid $N_2$, and then stored at −80°C for further use. Cell pellets were resuspended in 300 µl Breaking Buffer (100 mM Tris, 8.0, 20% glycerol, protease inhibitors [Millipore Sigma #04693159001]). Cells were lysed using 0.4 ml zirconia/silica beads (RPI #9834) in a Mini Beadbeater-96 (BioSpec Products) for 5 min. Lysates were spun at 21K × g for 2 min. Pellets were resuspended in 1 ml FA buffer (50 mM HEPES, 7.5, 150 mM NaCl, 1 mM EDTA, 1% Triton X-100, 0.1% sodium deoxycholate, protease inhibitors [Millipore Sigma #04693159001]) and transferred to 15 ml polystyrene tubes. In experiments with antibodies specific against phosphorylated Rpb1 CTD, Breaking Buffer and FA buffer were supplemented with phosphatase inhibitors (Thermo Fisher Scientific #A32957). Samples were sonicated in a cold Bioruptor sonicator bath (Diagenode #UCD-200) at a maximum output, cycling 30 s on, 30 s off, for a total of 45 min. Samples were spun twice in fresh tubes at 21K × g for 15 min. Prepared chromatin was flash-frozen in liquid $N_2$, and then stored at −80°C for further use.

20 µl of the chromatin sample was used to estimate DNA concentration. First, 20 µl Stop buffer (20 mM Tris, 8.0, 100 mM NaCl, 20 mM EDTA, 1% SDS) was added to samples followed by incubation at 70°C for 16–20 hr. Samples were digested with 0.5 mg/ml RNase A (Thermo Fisher Scientific #EN0531) for 30 min at 55°C and 1 mg/ml Proteinase K for 90 min at 55°C. Sample volume was brought to 200 µl, and DNA was purified by two phenol/CHCl₃/isoamyl alcohol (25:24:1) extractions and ethanol precipitation. DNA was resuspended in 20 µl 10 mM Tris, 8.0, and the concentration was measured using Qubit HS DNA assay (Thermo Fisher Scientific #Q32851).

20 µl Protein G Dynabeads (Thermo Fisher Scientific #10003D) was used for a single immunoprecipitation. Beads were first washed three times with 500 µl PBST buffer (PBS buffer supplemented with 0.1% Tween 20) for 3 min with gentle rotation. Beads were resuspended in a final volume of 20 µl containing PBST buffer and 5–8 µl of appropriate antibody (Key resources table). The following antibody volumes were used: H4K12ac, FLAG-Tag, Myc-Tag – 5 µl; Rpb1 CTD (total or phosphorylated) – 8.5 µl. The bead suspension was incubated for 60 min with shaking (1400 rpm) at RT, washed with 500 µl PBST buffer and 500 µl FA buffer. Beads were finally resuspended in 25 µl FA buffer. 1.5 µg *S. cerevisiae* chromatin and 30 ng *S. pombe* chromatin (strain SHY1110) were combined, and samples were brought to a final volume of 500 µl. 25 µl of each sample was mixed with 25 µl Stop buffer and set aside (input sample). 25 µl of beads was added to remaining 475 µl of samples followed by incubation for 16–20 hr at 4°C.

The beads were washed for 3 min with gentle rotation with the following: three times with 500 µl FA buffer, two times with FA-HS buffer (50 mM HEPES, 7.5, 500 mM NaCl, 1 mM EDTA, 1% Triton X-100, 0.1% sodium deoxycholate), and once with 500 µl RIPA buffer (10 mM Tris, 8.0, 0.25 M LiCl, 0.5% NP-40, 1 mM EDTA, 0.5% sodium deoxycholate). DNA was eluted from beads with 25 µl Stop buffer at 75°C for 10 min. Elution was repeated, eluates were combined and incubated at 70°C for 16–20 hr together with input samples collected earlier. Samples were digested with 0.5 mg/ml RNase A (Thermo Fisher Scientific #EN0531) for 30 min at 55°C and 1 mg/ml Proteinase K for 2 hr at 55°C. Sample volume was brought to 200 µl, and DNA was purified by two phenol/CHCl₃/isoamyl alcohol (25:24:1) extractions and ethanol precipitation. DNA was resuspended in 15 µl 10 mM Tris, 8.0, and the concentration was measured using Qubit HS DNA assay (Thermo Fisher Scientific #Q32851).

## Preparation of NGS libraries for ChEC-seq and ChIP-seq samples

NGS libraries for ChEC-seq and ChIP-seq experiments were prepared similarly as described (*Donczew et al., 2020*; *Warfield et al., 2017*). 12 µl of ChEC samples and 5 µl of ChIP samples were used as input for library preparation. Samples were end-repaired, phosphorylated, and adenylated in 50 µl reaction volume using the following final concentrations: 1X T4 DNA ligase buffer (NEB #B0202S), 0.5 mM each dNTP (Roche #KK1017), 0.25 mM ATP (NEB #P0756S), 2.5% PEG 4000, 2.5 U T4 PNK (NEB #M0201S), 0.05 U T4 DNA polymerase (Invitrogen #18005025), and 0.05 U Taq DNA polymerase (Thermo Fisher Scientific #EP0401). Reactions were incubated at 12°C 15 min, 37°C 15 min, 72°C 20 min, then put on ice and immediately used in adaptor ligation reactions. Adaptor ligation was performed in a 115 µl volume containing 6.5 nM adaptor, 1X Rapid DNA ligase buffer (Enzymatics #B101L) and 3000 U DNA ligase (Enzymatics #L6030-HC-L), and reactions were

incubated at 20° for 15 min. Following ligation, a two-step cleanup was performed for ChEC-seq samples using 0.25× vol Mag-Bind reagent (Omega Biotek # M1378-01) in the first step and 1.1× vol in the second step. In case of ChIP-seq samples, a single cleanup was performed using 0.4× vol Mag-Bind reagent. In both cases, DNA was eluted with 20 µl 10 mM Tris, 8.0. Library Enrichment was performed in a 30 µl reaction volume containing 20 µl DNA from the previous step and the following final concentrations: 1X KAPA buffer (Roche #KK2502), 0.3 mM each dNTP (Roche #KK1017), 2.0 µM each P5 and P7 PCR primer, and 1 U KAPA HS HIFI polymerase (#KK2502). DNA was amplified with the following program: 98℃ 45 s, (98℃ 15 s, ramp to 60℃ @ 3℃/s, 60℃ 10 s, ramp to 98℃ @ 3℃/s) 16–18×, 72℃ 1 min. 18 cycles were used for library amplification for ChEC-seq samples and 16 cycles for ChIP-samples. A post-PCR cleanup was performed using 1.4× vol Mag-Bind reagent, and DNA was eluted into 30 µl 10 mM Tris, 8.0. Libraries were sequenced on the Illumina HiSeq2500 platform using 25 bp paired-end reads at the Fred Hutchinson Cancer Research Center Genomics Shared Resources facility.

## Analysis of NGS data
Data analysis was performed similarly as described (*Donczew et al., 2020*). The majority of the data analysis tasks except sequence alignment, read counting, and peak calling (described below) were performed through interactive work in the Jupyter Notebook (https://jupyter.org) using Python programming language (https://www.python.org) and short Bash scripts. All figures were generated using Matplotlib and Seaborn libraries for Python; (https://matplotlib.org; https://seaborn.pydata.org). All code snippets and whole notebooks are available upon request.

Paired-end sequencing reads were aligned to *S. cerevisiae* reference genome (sacCer3), *S. pombe* reference genome (ASM294v2.20), or *D. melanogaster* reference genome (release 6.06) with Bowtie (*Langmead and Salzberg, 2012*) using optional arguments '-l 10 -X 700 `–local –very-sensitive-local –no-unal –no-mixed –no-discordant`'. Details of the analysis pipeline depending on the experimental technique used are described below.

## Analysis of RNA-seq data
SAM files for *S. cerevisiae* data were used as an input for HTseq-count (*Anders et al., 2015*) with default settings. The GFF file with *S. cerevisiae* genomic features was downloaded from the Ensembl website (assembly R64-1-1). Signal per gene was normalized by the number of all *S. pombe* reads mapped for the sample and multiplied by 10,000 (arbitrarily chosen number). Genes classified as dubious, pseudogenes, or transposable elements were excluded, leaving 5797 genes for the downstream analysis. As a next filtering step, we excluded all the genes that had no measurable signal in at least one out of 42 samples collected in this work. The remaining 5313 genes were used to calculate coefficient of variation (CV) to validate reproducibility between replicate experiments (*Figure 1—figure supplement 1D*, *Figure 3—figure supplement 1B* and *Supplementary file 1*). This gene set was further compared to a list of 4900 genes we found previously to provide the best reproducibility with a minimal loss of information (*Donczew et al., 2020*). The overlapping set of 4883 genes was used in all plots where genes were divided into TFIID-dependent and CR categories. The results of biological replicate experiments for each sample were averaged. Corresponding samples were compared to calculate $\log_2$ change in expression per gene (IAA to DMSO samples for degron experiments, and *BDF2* deletion mutant to WT strain [BY4705]) (*Supplementary file 1*).

## Analysis of ChEC-seq data
SAM files for *S. cerevisiae* data were converted to tag directories with the HOMER (http://homer.ucsd.edu; *Heinz et al., 2010*) 'makeTagDirectory' tool. Data for six DMSO-treated replicate samples from experiments involving Bdf1, Bdf2, or Bdf1/2 depletion were used to define promoters bound by Bdf1, Bdf2, Taf1, Taf11, Med8, Med17, and Spt3 as described below (*Supplementary file 3*). Peaks were called using HOMER 'findPeaks' tool with optional arguments set to '-o auto -C 0 L 2 F 2', with the free MNase dataset used as a control. These settings use a default false discovery rate (FDR; 0.1%) and require peaks to be enriched twofold over the control and twofold over the local background. Resulting peak files were converted to BED files using 'pos2bed.pl' program. For each peak, the peak summit was calculated as a mid-range between peak borders. For peak assignment to promoters, the list of all annotated ORF sequences (excluding sequences classified as 'dubious'

or 'pseudogene') was downloaded from the SGD website (https://www.yeastgenome.org). Data for 5888 genes were merged with TSS positions (*Park et al., 2014*). If the TSS annotation was missing, TSS was manually assigned at position −100 bp relative to the start codon. Peaks were assigned to promoters if their peak summit was in the range from −300 to +100 bp relative to TSS. In a few cases, where more than one peak was assigned to the particular promoter, the one closer to TSS was used. Promoters bound in at least four out of six replicate experiments were included in a final list of promoters bound by a given factor and were used to calculate promoter occupancy in all relevant experiments.

Coverage at each base pair of the *S. cerevisiae* genome was calculated as the number of reads that mapped at that position divided by the number of all *D. melanogaster* reads mapped for the sample and multiplied by 10,000 (arbitrarily chosen number). To quantify $\log_2$ change in factor promoter, occupancy signal per promoter was calculated as a sum of normalized reads per base in a 200 bp window around the promoter peak summit. Peak summits were defined using HOMER as described above. If no peak was assigned to a promoter using HOMER, the position of the strongest signal around TSS was used as a peak summit. Manual inspection of selected cases confirmed the validity of this approach. Corresponding IAA and DMSO-treated samples were compared to calculate $\log_2$ change in occupancy, and the results of biological replicate experiments for each sample were averaged (*Supplementary file 3*).

## Analysis of ChIP-seq data

Data for Bdf1 was used to call peaks as described for ChEC-seq with input sample used as a control. Promoters bound in at least one out of two replicate experiments were included in a final list of Bdf1 bound promoters as defined by ChIP-seq. For all samples, coverage at each base pair of the *S. cerevisiae* genome was calculated as the number of reads that mapped at that position divided by the number of all *S. pombe* reads mapped in the sample, multiplied by the ratio of *S. pombe* to *S. cerevisiae* reads in the corresponding input sample and multiplied by 10,000 (arbitrarily chosen number). The list of all annotated ORF sequences (excluding sequences classified as 'dubious' or 'pseudogene') was downloaded from the SGD website (https://www.yeastgenome.org). Data for 5888 genes were merged with TSS positions obtained from *Park et al., 2014*. If the TSS annotation was missing, the TSS was manually assigned at position −100 bp relative to the start codon. Signal per gene was calculated as a sum of normalized reads per base in a fixed window relative to TSS (defined in figure legends). The results of biological duplicate experiments for each sample were averaged and the $\log_2$ change in signal per gene was calculated by comparing corresponding IAA and DMSO-treated samples (*Supplementary file 3*). In experiments mapping Rpb1 CTD, Bur1, Ctk1, and Spt5, the signal per gene was calculated between TSS and PAS (*Park et al., 2014*; *Supplementary file 4*).

FASTQ files for the Bdf1 ChIP-exo (SRR397550) and H3 ChIP-seq (SRR6495880, SRR6495888, SRR6495913, SRR6495921) experiments were obtained from the SRA. Data were processed as described above except for the use of RPM normalization. The four datasets for H3 were averaged and used to normalize H4K12ac data.

## Phylogenetic analysis

Amino acid sequences of yeast and human BET proteins and human Taf1 were downloaded from NCBI RefSeq database. Positions of individual BDs were obtained from *Wu and Chiang, 2007*. Multiple sequence alignment of individual BDs was performed using Clustal Omega (*Sievers et al., 2011*). Neighbor-joining tree was visualized with Jalview v 2.11.1.3 using PAM250 scoring matrix (*Waterhouse et al., 2009*).

## Gene ontology analysis

Gene ontology analysis for yeast and human BET regulated genes was done using PANTHER over-representation test and PANTHER GO-Slim Biological Process annotation dataset with default settings (http://www.pantherdb.org; v 16.0) (*Mi et al., 2021*). Only the most specific, enriched child processes with FDR < 0.05 were used to compare BET functions between yeast and human model systems.

## Acknowledgements

We thank Christine Cucinotta and Sarah Swygert for helpful comments about ChIP-seq experiments, Toshi Tsukiyama for plasmids, and Sandipan Brahma and all members of the Hahn lab for comments on manuscript. This work was supported by grants NIH GM053451, GM075114 and GM140823 to SH and NIH P30 CA015704 to the Fred Hutch Genomics and Computational Shared Resources facility.

## Additional information

### Funding

| Funder | Grant reference number | Author |
|---|---|---|
| National Institute of General Medical Sciences | RO1GM053451 | Steven Hahn |
| National Institute of General Medical Sciences | RO1GM075114 | Steven Hahn |
| National Institute of General Medical Sciences | R35GM140823 | Steven Hahn |

The funders had no role in study design, data collection and interpretation, or the decision to submit the work for publication.

### Author contributions

Rafal Donczew, Conceptualization, Resources, Data curation, Formal analysis, Validation, Investigation, Visualization, Methodology, Writing - original draft, Writing - review and editing; Steven Hahn, Conceptualization, Resources, Supervision, Funding acquisition, Project administration, Writing - review and editing

### Author ORCIDs

Rafal Donczew (ID) https://orcid.org/0000-0001-9729-4153
Steven Hahn (ID) https://orcid.org/0000-0001-7240-2533

### Decision letter and Author response

Decision letter https://doi.org/10.7554/eLife.69619.sa1
Author response https://doi.org/10.7554/eLife.69619.sa2

## Additional files

### Supplementary files

• Supplementary file 1. Summary of 4-thioU RNA-seq experiments. Related to *Figures 1* and *3*.

• Supplementary file 2. Results of gene ontology analysis on 25% most bromodomain and extra-terminal domain (BET)-sensitive genes in yeast (this work) and human (*Winter et al., 2017*) system. Related to *Figure 1*.

• Supplementary file 3. Summary of ChEC-seq and ChIP-seq experiments. Related to *Figures 2–5*.

• Supplementary file 4. Summary of ChIP-seq experiments on Rpb1, Ctk1, Bur1, and Spt5. Related to *Figure 6*.

• Supplementary file 5. Yeast strains used in this study.

• Supplementary file 6. Number of biological replicates collected for all experiments in this study.

• Transparent reporting form

## Data availability

The datasets generated during this study are available at Gene Expression Omnibus under accession GSE171067.

The following dataset was generated:

| Author(s) | Year | Dataset title | Dataset URL | Database and Identifier |
|---|---|---|---|---|
| Donczew R, Hahn S | 2021 | BET family members Bdf1/2 modulate global transcription initiation and elongation in Saccharomyces cerevisiae | https://www.ncbi.nlm.nih.gov/geo/query/acc.cgi?acc=GSE171067 | NCBI Gene Expression Omnibus, GSE171067 |

The following previously published datasets were used:

| Author(s) | Year | Dataset title | Dataset URL | Database and Identifier |
|---|---|---|---|---|
| Winter GE | 2017 | BET bromodomain proteins function as master transcription elongation factors independent of CDK9 recruitment | https://www.ncbi.nlm.nih.gov/geo/query/acc.cgi?acc=GSE79290 | NCBI Gene Expression Omnibus, GSE79290 |
| Donczew R | 2020 | Two separate roles for the transcription coactivator SAGA and a set of genes redundantly regulated by TFIID and SAGA | https://www.ncbi.nlm.nih.gov/geo/query/acc.cgi?acc=GSE142122 | NCBI Gene Expression Omnibus, GSE142122 |
| Bruzzone MJ | 2018 | Distinct patterns of histone acetyltransferase and Mediator deployment at yeast protein-coding genes | https://www.ncbi.nlm.nih.gov/geo/query/acc.cgi?acc=GSE109235 | NCBI Gene Expression Omnibus, GSE109235 |
| Oberbeckmann E | 2019 | Absolute nucleosome occupancy map for the Saccharomyces cerevisiae genome | https://www.ncbi.nlm.nih.gov/geo/query/acc.cgi?acc=GSE132225 | NCBI Gene Expression Omnibus, GSE132225 |

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
