## [Decision Letter]

**Acceptance summary:**

Donczew and Hahn provide extensive genomic characterization of the global roles of the bromodomain-containing factors Bdf1 and Bdf2 in the yeast *Saccharomyces cerevisiae*. The data help to resolve longstanding questions about the roles of bromodomains as readers of acetylation marks, and raise new questions about how bromodomain-containing proteins support the functions of TFIID and other components of the transcription machinery in acetylation-independent ways. The authors argue that Bdf1/2 in yeast are more similar in sequence and function to mammalian Brd4 than was previously assumed. Notably, this includes a role in early transcript elongation 300-400 bp after initiation that may be part of an elongation checkpoint. This work challenges our understanding of the functions of bromodomain-containing proteins in yeast and higher organisms, and opens the use of the yeast system to investigate these questions.

**Decision letter after peer review:**

Thank you for submitting your article "BET family members Bdf1/2 modulate global transcription initiation and elongation in *Saccharomyces cerevisiae*" for consideration by *eLife*. Your article has been reviewed by 3 peer reviewers, including Tim Formosa as the Reviewing Editor and Reviewer #1, and the evaluation has been overseen by Jessica Tyler as the Senior Editor.

Essential revisions:

1) The authors should clarify what is meant by "processive elongation" and how their data support this key conclusion. The data in Figure 6B indicate decreasing RNA Pol II occupancy over about the first 300 bp in 3 of the 5 classes of genes, but then the effect plateaus. This observation does not fit well with either a role in release of pausing (expected to be more limited to the region near the TSS) or general processivity (expected to occur across the entire length of the genes). The traveling ratio does not seem well suited to measure this as it only takes account of the initial and final occupancies, but the authors have the data needed to ask about processivity across the genes that could answer this question more completely.

2) What is the relative contribution of Bdf1 and Bdf2 to the elongation function?

3) Do Bdf1 and Bdf2 effects on initiation correlate with the effect on elongation at the same genes? This is alluded to in the Discussion but not analyzed in the Results.

4) The change in transcription noted upon Brd1/2 depletion in Figure S3C is surprisingly similar for genes with and without scored Bdf1 peaks. While the difference is noted to be statistically significant, the absolute change does not appear to be likely to be physiologically important. Do the authors think this means there are indirect effects of Brd1/2 on all genes, that there is a problem comparing the 4-thioU-seq data with CHEC-seq data, or do they have some other explanation? Perhaps comparing continuous occupancy values (the data in Figure S3D, for example) instead of binary peak calls would resolve this puzzling observation?

5) Similarly, Figure 2D shows statistical significance for the difference in histone acetylation for genes with and without Bdf1 peak calls, but the absolute difference seems small. The authors should discuss why histone acetylation is less of a driver of occupancy than expected for bromodomain factors.

6) The conclusion that Bdf1/2 have roles independent of TFIID relies on similar kinetics and extents of depletion of the Bdfs and Taf1/Taf13 but this is not shown here.

7) The authors interpret Ser2 phosphorylation as a marker for release of pausing, but the roles of P-TEFb on DSIF, NELF, and other factors seem more firmly established, so this interpretation needs to be explained more completely. Perhaps related to this and to point 1, the authors should also discuss the increased occupancy by Ctk1 and Bur1, as this might indicate failure to perform a transition between initiation and elongation, leading to elevated accumulation of the enzymes that participate in promoting it. Does the primary data suggest accumulation of Ctk1 and Bur1 over the promoter as seen with RNA Pol II, or is it found throughout the gene bodies?

*Reviewer #1:*

Histone acetylation correlates with active transcription, and can be read by the bromodomain sequence motifs found in several proteins, but the connection between the readers and functional outcomes remains unclear. TFIID is an important part of the transcription machinery in all eukaryotes, but while the version in higher eukaryotes contains bromodomains, these are missing from the yeast factor, with this activity presumed to be supplied by two functionally overlapping factors, Bdf1 and Bdf2. This manuscript investigates the roles of these factors in yeast using a range of genomic approaches, finding some support for the expected functions in cooperation with TFIID but also showing that the importance of Bdf1 and Bdf2 overlaps but exceeds those of factors involved in histone acetylation, TFIID action, and pre-initiation complex (PIC) formation. The authors argue that their results are more consistent with functions of the BET bromodomain-containing family in higher eukaryotes, especially Brd4, including roles in both transcription initiation and elongation. This opens the possibility of using the robust tools available in the yeast system to gain deeper understanding of a family of factors that is currently a target of cancer therapeutics. The role of Bdf1/2 in transcript elongation is key to this conclusion, but this part of the manuscript is somewhat underdeveloped. The effect on elongation noted could indicate a role in releasing RNA polymerase from pausing near the site of initiation or a role in enhancing processivity of elongation complex throughout transcription units, but the data don't fit either model well and the authors do not provide adequate interpretation of their existing results to help the reader understand their proposed model. Overall, the work has high significance in resolving longstanding issues regarding the functional roles of an important, conserved set of factors, it provides a set of valuable resources for the field, and it opens new avenues for using the yeast model to investigate a clinically relevant drug target in humans. Further clarity regarding the role in transcript elongation would strengthen the manuscript.

Key strengths:

The work shows:

• Distinct classes of effects of acute withdrawal of Bdf1/2, with higher impact on TFIID-dependent ("TATA-less") genes than CR ("TATA-dependent") genes.

• Similarity at the sequence level is higher for BETs than for Taf1

• Bdf1/2 localize to promoters, more in TFIID-dependent than in CR, and the change in Bdf1/2 occupancy correlates moderately with change in transcription (r = -0.52).

• Bromodomains suggest localization is driven by histone acetylation, and Esa1 depletion shows that Bdf1/2 depletion affects transcription of many of the same genes (r = 0.73). However, the Bdf1/2 depletion has larger effects than loss of histone acetylation, and loss of acetylation leads to diminished Bdf1/2 localization but with the same pattern of promoter enhancement, suggesting there are other ways to be localized and contribute to transcription.

• A role in PIC formation is strong, with Bdf1/2 depletion leading essentially to full loss of TFIIB localization to TSSs. Again, effects are stronger than depletion of TFIIB alone, so Bdf1/2 have additional roles beyond PIC formation.

Key issue to resolve:

The data show a decrease in RNA Pol II occupancy near the TSS upon Bdf1/2 depletion in the Bdf1/2-independent genes, and accumulation of RNA Pol II over the TSS in Bdf1/2-dependent genes. This seems consistent with a defect in promoter release at Bdf1/2-dependent genes, but it is less clear why this is considered to be a defect in "processive elongation." A decrease in elongation efficiency, especially reduced processivity, would be expected to cause a gradient of diminished RNA Pol II occupancy across gene bodies (high at the 5' end, diminishing across the gene as RNA Pol II dissociates more frequently than normal), but the observation seems to better support a defect just in the transition between initiation and elongation. This would also be consistent with the whole-gene decrease in S2 and S5 phosphorylation noted. Perhaps this could also explain the otherwise puzzling increased occupancy by Ctk1 and Bur1, as this could indicate failure to perform a transition function, leading to elevated accumulation of the enzymes that participate in promoting it. Does the primary data suggest accumulation of Ctk1 and Bur1 over the promoter as seen with RNA Pol II, or is it found throughout the gene bodies? A role in promoter release also seems more in line with the results with human BET factors. The authors should therefore clarify what they mean by "processive elongation" and describe more explicitly how they think the data (especially the results in Figure S7) support their conclusions.

Data availability and other issues are excellent.

*Reviewer #2:*

Yeast has two BET proteins, Bdf1 and Bdf2, that were previously proposed to provide bromodomains (BD) to TFIID. Indeed, yeast TFIID, unlike its mammalian counterparts, is lacking BD. Donczew and Hahn show that the BDs of Bdf1 and Bdf2 are more closely related to the BDs in Brd4 than those found in human TFIID. Furthermore, they provide experimental evidence suggesting that yeast Bdfs play similar roles to Brd4 in transcription. As for Brd4 in mammalian cells, Bdfs contribute to both PIC assembly (with a greater contribution at TATA-less promoters) and transcription elongation. This work opens the way to the use of the power of yeast genetics for understanding Brd4 (via Bdf1 and 2). The work would benefit from a deeper analysis of the role of Bdfs in elongation.

The most novel/impactful aspect of this work resides in the demonstration that Bdfs affect elongation in addition to initiation, hence suggesting they are functionally more related to Brd4 than to mammalian BD-containing TAFs. Yet, only one figure (Figure 6) is dedicated to this aspect. The authors describe in length the role of Bdfs in initiation, notably by dissecting what happens at different gene classes but only superficially characterize the mechanism behind the elongation function. Using their current data, the authors could/should easily address a few key questions, notably: 1-What is the relative contribution of Bdf1 vs Bdf2 to the elongation function; 2- Is there a link between the initiation and elongation functions with regards to the genes involved (is the dependence on Bdfs for elongation also biases against CR/TATAbox genes). This last point is eluded to in the Discussion (lines 409-420) but none of this is actually shown in the Result section. I am also concern about the sentence starting at line 300: "At many metazoan promoters, Pol II pauses after transcribing ~20-100 nucleotides and release of paused Pol II, mediated in part by the C-terminal repeat domain (CTD) Ser2 phosphorylation, is a critical step in gene regulation (Core and Adelman, 2019)". The evidence for the role of CTD Ser2-P in pause release is very thin. It is generally more accepted that the effect of P-TEFb on pause release is mediated by its phosphorylation of DSIF (Spt5), NELF, and perhaps other factors.

*Reviewer #3:*

The largest subunit of TFIID, Taf1, contains two bromodomains that show specificity for acetylated histone H4. The homologous protein in *S. cerevisiae* lacks these domains and a several decades-old study suggested that the role of the missing domains is performed by two redundant proteins, Bdf1 and Bdf2. In this study, the authors use genome-wide analyses to test this widely accepted model. The data show that Bdf1 and 2 are critical for transcription of a subset of genes, where they colocalize with TFIID. Additionally, BDF1/2- dependent genes are also dependent on ESA1, which encodes a histone H4-specific acetyltransferase. Interestingly, depletion of Bdf1 and 2 has more severe effects on gene expression than a loss of TFIID components suggested TFIID-independent roles for these proteins. Also surprising, the partial loss of TFIID or PIC components following Bdf1/2 depletion is insufficient to explain the large decrease in transcription observed upon loss of these proteins. Instead, the authors provide evidence that Bdf1 and 2 play roles in mediator recruitment and transcription elongation, which are hallmark roles for BRD4, a well-studied mammalian protein with roles in liquid-liquid phase separation and a target for anti-cancer therapeutics. This is supported by increased sequence similarity between the Bdf1/2 bromodomains and BRD4 compare to Taf1. Collectively, this work provides some support for existing models, implies that Bdf1 and 2 functions are more nuanced than previously thought, and suggests that these proteins are potential models for the study of BRD4.

1. To make the claim that Bdf1/2 have TFIID-independent roles the authors need to show that Taf1 and Taf13 are depleted with similar kinetics and extents as Bdf1 and 2.

2. Figure S3C shows that genes lacking a Bdf1 peak are equally dependent on Bdf1/2. Does this suggest that Bdf1/2 have an indirect function in some genes? Alternatively, is there a chance that CHEC-seq and 4-thioU RNA-seq should not be compared due to the impact of the different cell treatments on the transcription programs of the cells?

3. Figure 2D shows that for both TFIID and CR classes of genes, the normalized H4K12ac level at promoters significantly differs depending on the presence of a Bdf1 peak, but the difference is modest. Is this difference biologically meaningful? This modest difference is even more surprising when one considers the correlation between loss of Bdf1 and transcription following Esa1 depletion suggesting that the effect of Esa1 on transcription is largely mediated by Bdf1 (Figure 3E). Why aren't all acetylated regions bound by Bdf1?

---

## [Author Response]

Essential revisions:1) The authors should clarify what is meant by "processive elongation" and how their data support this key conclusion. The data in Figure 6B indicate decreasing RNA Pol II occupancy over about the first 300 bp in 3 of the 5 classes of genes, but then the effect plateaus. This observation does not fit well with either a role in release of pausing (expected to be more limited to the region near the TSS) or general processivity (expected to occur across the entire length of the genes). The traveling ratio does not seem well suited to measure this as it only takes account of the initial and final occupancies, but the authors have the data needed to ask about processivity across the genes that could answer this question more completely.

Thank you for the insightful comments. The following answer also relates to point 7:

We agree that the term 'processive elongation' was not a good description of the defects in transcription elongation caused by Bdf depletion. We changed it to 'early elongation' since the detected changes occur in the first 300-400 bp from TSS. As noted by the Reviewers, this observation does not fit with the pause release model observed in metazoans or with general changes in Pol II processivity. Interestingly, the Pugh lab recently reported that acute stress in *S. cerevisiae* causes Pol II stalling (Badjatia et al., 2021) at the +2 nucleosome, and suggested a regulatory elongation checkpoint. This stalling region overlaps the region where we find elongation defects in the absence of Bdfs. This connection is now discussed on lines 463-466, pg 14.

We performed additional analysis of the data collected for Bur1, Ctk1 and Spt5. We found that the major accumulation of Bur1 and Ctk1 occurs over the first ~400 bp and anti-correlates with changes in Pol II, i.e., most Bdf-dependent genes experience the biggest loss of Pol II and the biggest increase in relative Bur1 and Ctk1 occupancy (Figure 6G). This result is counterintuitive considering the loss of Ser2 phosphorylation. Human BET factors Brd2 and Brd4 were claimed to have kinase activity and Brd4 was directly connected to Ser2 phosphorylation independently of P-TEFb (Denis and Green, 1996; Devaiah et al., 2012). Similarly, Bdf1 was also reported to have an intrinsic kinase activity (Matangkasombut et al., 2000). This possible Bdf1 kinase activity would explain the observed defects and we plan to investigate it in the future work.

At this point, we don't know what drives the increase in Bur1 and Ctk1 occupancy in relation to the Rpb1. We introduced a separate paragraph to the Results section (pg 11) where results related to Bur1, Ctk1 and Spt5 are presented (lines 374-397). Relevant changes were also made in the Discussion (lines 478-487, pg 15).

Finally, we believe that the travelling ratio calculated as the ratio of Pol II at TSS and TES is a useful metric to quantify the observed changes in Pol II distributions at individual genes. To validate our approach, we calculated Pol II signal at other locations along the transcribed regions and related it to signal at TSS. Starting at ~400 bp from TSS the results were very similar and supported our conclusions based on calculation at TES.

2) What is the relative contribution of Bdf1 and Bdf2 to the elongation function?

In the presence of Bdf1, the contribution of Bdf2 to transcription is minimal as shown in Figure 1. Depletion of Bdf1 in the presence of Bdf2 gives a modest defect in transcription and only the simultaneous depletion of Bdf1/2 causes a strong defect. For these reasons all experiments using ChEC-seq and ChIP-seq were based on depletion of both Bdf factors to maximize the observed defects. We expect that, similarly to results measured by 4-thioU RNA-seq, the contribution of Bdf2 to elongation function is small and it would be hard to capture differences in Rpb1 distribution if the Pol II ChIP-seq experiment was based on only Bdf2 depletion.

3) Do Bdf1 and Bdf2 effects on initiation correlate with the effect on elongation at the same genes? This is alluded to in the Discussion but not analyzed in the Results.

To clarify this point, we added Figure 6—figure supplement 2C, which compares the change in TFIIB binding with the change in travelling ratio. The correlation is weak and suggests that, at many genes, the relative contributions of Bdf1/2 to initiation and elongation can be significantly different. Corresponding changes were made in the manuscript (lines 352-353; pg 11).

4) The change in transcription noted upon Brd1/2 depletion in Figure S3C is surprisingly similar for genes with and without scored Bdf1 peaks. While the difference is noted to be statistically significant, the absolute change does not appear to be likely to be physiologically important. Do the authors think this means there are indirect effects of Brd1/2 on all genes, that there is a problem comparing the 4-thioU-seq data with CHEC-seq data, or do they have some other explanation? Perhaps comparing continuous occupancy values (the data in Figure S3D, for example) instead of binary peak calls would resolve this puzzling observation?

Thank you for the comment. Following the suggestion, we replaced Figure S3D with a scatter plot (now Figure 2—figure supplement 1C), which compares the loss of transcription after depleting Bdf1/2 with the Bdf1 promoter signal calculated in a fixed window for all 4883 genes analyzed by RNA-seq. The gene dependence on Bdfs shows a relatively weak correlation with Bdf1 promoter occupancy. Similar findings were reported for the human BET factors. In Winter et al., (2017) BRD4 enhancer occupancy has limited capacity to predict gene response to BET degradation. Similarly, Muhar et al. (2018), found that BRD4 occupancy did not predict gene response to JQ1 treatment. Corresponding changes were made in the manuscript (lines 211-214; p7).

5) Similarly, Figure 2D shows statistical significance for the difference in histone acetylation for genes with and without Bdf1 peak calls, but the absolute difference seems small. The authors should discuss why histone acetylation is less of a driver of occupancy than expected for bromodomain factors.

Thank you for the comment. We modified the manuscript in lines 269 – 274 (pg 9) where we discuss possible alternative targeting mechanism for Bdfs including interactions of Bdf1 with acetyl lysines on histone H3 and reference examples of human BET factors interacting with other transcription factors in a bromodomain-independent manner. For example, the conserved ET domain seems to be a frequent point of contact with protein partners. It is possible that the Bdf1 ET domain is similarly involved in interactions with non-histones proteins and such interactions can provide alternative means of recruitment of Bdf1 to chromatin.

6) The conclusion that Bdf1/2 have roles independent of TFIID relies on similar kinetics and extents of depletion of the Bdfs and Taf1/Taf13 but this is not shown here.

Thank you for the suggestion. We compared the kinetics of Bdf1 and Taf1/Taf13 degradation by Western blot. The results shown in Figure 1—figure supplement 2A validate that the degradation of Bdf1 (both alone or in combination with *bdf2* deletion) is comparable to Taf1 and Taf13 degradation at time points before, during and after the 30 minute depletion time used for our RNA analysis. Corresponding change was made in the manuscript (line 142-143; pg 5).

7) The authors interpret Ser2 phosphorylation as a marker for release of pausing, but the roles of P-TEFb on DSIF, NELF, and other factors seem more firmly established, so this interpretation needs to be explained more completely. Perhaps related to this and to point 1, the authors should also discuss the increased occupancy by Ctk1 and Bur1, as this might indicate failure to perform a transition between initiation and elongation, leading to elevated accumulation of the enzymes that participate in promoting it. Does the primary data suggest accumulation of Ctk1 and Bur1 over the promoter as seen with RNA Pol II, or is it found throughout the gene bodies?

Thank you for the comments. We changed the manuscript to include information about phosphorylation of NELF and DSIF. Our answer to the question regarding Bur1 and Ctk1 is included in the response to the point 1 above and includes additional data analysis now shown in Figure 6G and in Figure 6—figure supplement 3.